# The enteric nervous system of the *C. elegans* pharynx is specified by the Sine oculis-like homeobox gene *ceh-34*

**Berta Vidal, Burcu Gulez, Wen Xi Cao, Eduardo Leyva-Díaz, Molly B Reilly, Tessa Tekieli, Oliver Hobert***

Department of Biological Sciences, Columbia University, Howard Hughes Medical Institute, New York, United States

**Abstract** Overarching themes in the terminal differentiation of the enteric nervous system, an autonomously acting unit of animal nervous systems, have so far eluded discovery. We describe here the overall regulatory logic of enteric nervous system differentiation of the nematode *Caenorhabditis elegans* that resides within the foregut (pharynx) of the worm. A *C. elegans* homolog of the *Drosophila* Sine oculis homeobox gene, *ceh-34*, is expressed in all 14 classes of interconnected pharyngeal neurons from their birth throughout their life time, but in no other neuron type of the entire animal. Constitutive and temporally controlled *ceh-34* removal shows that *ceh-34* is required to initiate and maintain the neuron type-specific terminal differentiation program of all pharyngeal neuron classes, including their circuit assembly. Through additional genetic loss of function analysis, we show that within each pharyngeal neuron class, *ceh-34* cooperates with different homeodomain transcription factors to individuate distinct pharyngeal neuron classes. Our analysis underscores the critical role of homeobox genes in neuronal identity specification and links them to the control of neuronal circuit assembly of the enteric nervous system. Together with the pharyngeal nervous system simplicity as well as its specification by a Sine oculis homolog, our findings invite speculations about the early evolution of nervous systems.

**\*For correspondence:**
or38@columbia.edu

**Competing interest:** The authors declare that no competing interests exist.

## Editor's evaluation

This paper marks a significant advance in understanding the transcriptional control of neural cell fate and connectivity. The authors show that a single homeodomain transcription factor has a central role in specifying the diverse neuron types of the enteric nervous system of the *C. elegans* pharynx. By linking cell fates across a single, largely self-contained circuit, these studies support the emerging idea that transcriptional control can link cell fate to circuit connectivity and function.

## Introduction

Across animal phylogeny, enteric nervous systems constitute a self-contained, autonomously acting neuronal network that detects physiological conditions to control the peristaltic movement of food through the digestive tract (*Ayali, 2004*; *Copenhaver, 2007*; *Fung and Vanden Berghe, 2020*; *Hartenstein, 1997*; *Laranjeira and Pachnis, 2009*). Because of structural and functional autonomy, the enteric nervous system has been referred to as a 'second brain' (*Gershon, 1998*). In mammals, the enteric nervous system is composed of around 20 different neuron types, categorized into intrinsic sensory, inter- or motor neurons, which line the interior lumen of different sections of the digestive system (*Drokhlyansky et al., 2020*; *Furness, 2000*). While progress has been made in understanding early developmental patterning events that establish the fate of neurons in the enteric nervous system

of mammals (**Nagy and Goldstein, 2017**; **Sasselli et al., 2012**), fish (**Ganz, 2018**), and flies (**Copenhaver, 2007**; **Myers et al., 2018**), much less is known about terminal differentiation programs of enteric neurons, both in vertebrate and invertebrate models (**Copenhaver, 2007**; **Hao and Young, 2009**; **Memic et al., 2018**; **Morarach et al., 2021**; **Rao and Gershon, 2018**). Specifically, it has remained unclear as to whether there are common unifying themes in how enteric neurons acquire their terminally differentiated state. This is particularly interesting from an evolutionary standpoint. The function of enteric neurons in controlling feeding behavior is an ancient one that may precede the evolution of the bilaterian central nervous system (**Cook et al., 2020**; **Furness and Stebbing, 2018**; **Gilbert, 2019**; **Koizumi, 2007**). Understanding how enteric neurons acquire their terminal features may therefore provide novel insights into nervous system evolution.

The nematode *Caenorhabditis elegans* contains an autonomously acting nervous system in its foregut, the pharynx, composed of 20 synaptically interconnected neurons that fall into 14 anatomically distinct classes (**Albertson and Thomson, 1976**; **Cook et al., 2020**; **Mango, 2007**; **Figure 1**). Due to its association with the digestive tract of the worm, the pharyngeal nervous system can be considered to be the enteric nervous system of *C. elegans*. Apart from this anatomical association, the pharyngeal nervous system shares functional features of enteric nervous systems of more complex animals. It is required for movement of food through the digestive tract of the worm and functions in an entirely autonomous manner, even if removed from the rest of the animal (**Albertson and Thomson, 1976**; **Avery, 2012**; **Cook et al., 2020**; **Mango, 2007**). Like other enteric nervous systems, the nematode pharyngeal nervous system constitutes a non-centralized neuronal network isolated from the rest of the nervous system and, in rough analogy to the vagus nerve, is connected to the remainder of the nervous system through a single nerve fiber, that of the bilateral RIP neuron pair (**Albertson and Thomson, 1976**; **Cook et al., 2020**; **Cook et al., 2019**; **White et al., 1986**). Like vertebrate enteric neurons (**Fung and Vanden Berghe, 2020**), pharyngeal neurons have sensory, inter- and motorneuron function (**Avery and Horvitz, 1989**; **Cook et al., 2020**; **Trojanowski et al., 2014**).

Our recent re-analysis of pharyngeal nervous system anatomy has shown that rather than segregating these functions over distinct neurons as vertebrates do (**Fung and Vanden Berghe, 2020**), most pharyngeal neurons each combine sensory, inter- and motorneuron function, that is, are polymodal (**Cook et al., 2020**). The 14 distinct pharyngeal neuron types are defined by their unique anatomy, that is, axonal projections, morphology, synaptic connectivity (**Figure 1B and C**; **Cook et al., 2020**), and unique functional features (**Avery and Horvitz, 1989**; **Trojanowski et al., 2014**). This anatomical and functional classification has recently been further extended by the description of their unique molecular fingerprint, determined by scRNA transcriptomic analysis (**Taylor et al., 2021**; **Figure 1D**). For example, each pharyngeal neuron class is uniquely defined by characteristic signatures of neurotransmitter systems, from acetylcholine (ACh), glutamate (Glu) to serotonin, and by unique combinations of neuropeptide-encoding genes (**Figure 1E**). While pharyngeal neurons are clearly different from one another, scRNA profiling has shown that their molecular signatures are more similar to each other than to other neurons in the nervous system (**Taylor et al., 2021**; **Figure 1D**).

One fascinating aspect of the *C. elegans* pharyngeal nervous system is that it has an appearance of what one could imagine an ancestral, primitive nervous system to have looked like. Primitive nervous systems are generally thought to have emerged in the context of monolayers of epithelial cells, with individual cells in such layers specializing into primitive sensory motor-type neurons (**Arendt, 2008**; **Mackie, 1970**; **Varoqueaux and Fasshauer, 2017**). These diffusely organized neurons may have sensed the environment and relayed such sensory information to the other primitive cell type thought to have arisen early in evolution, namely, contractile 'myoepithelial' cells that were able to generate motion. The organization of the *C. elegans* pharynx reveals some striking parallels to such a presumptive primitive nervous system: It is also essentially a single monolayer of cells that is organized into a tubular structure and the vast majority of constituent cells are myoepithelial cells (pharyngeal muscle) and polymodal, interconnected sensory/motor neurons (**Cook et al., 2020**; **Mango, 2007**; **Portereiko and Mango, 2001**). Pharyngeal neurons combine sensory, inter- and motor neuron features and are also not localized to ganglia but rather diffusely localized, resembling the architecture of more ancient nerve nets (**Albertson and Thomson, 1976**; **Watanabe et al., 2009**). The interconnectivity of pharyngeal neurons also displays less selectivity than non-pharyngeal neurons, such that mere physical proximity is an almost sufficient criterion for connectivity (**Cook et al., 2020**). Moreover, pharyngeal neurons are closely related to non-neuronal

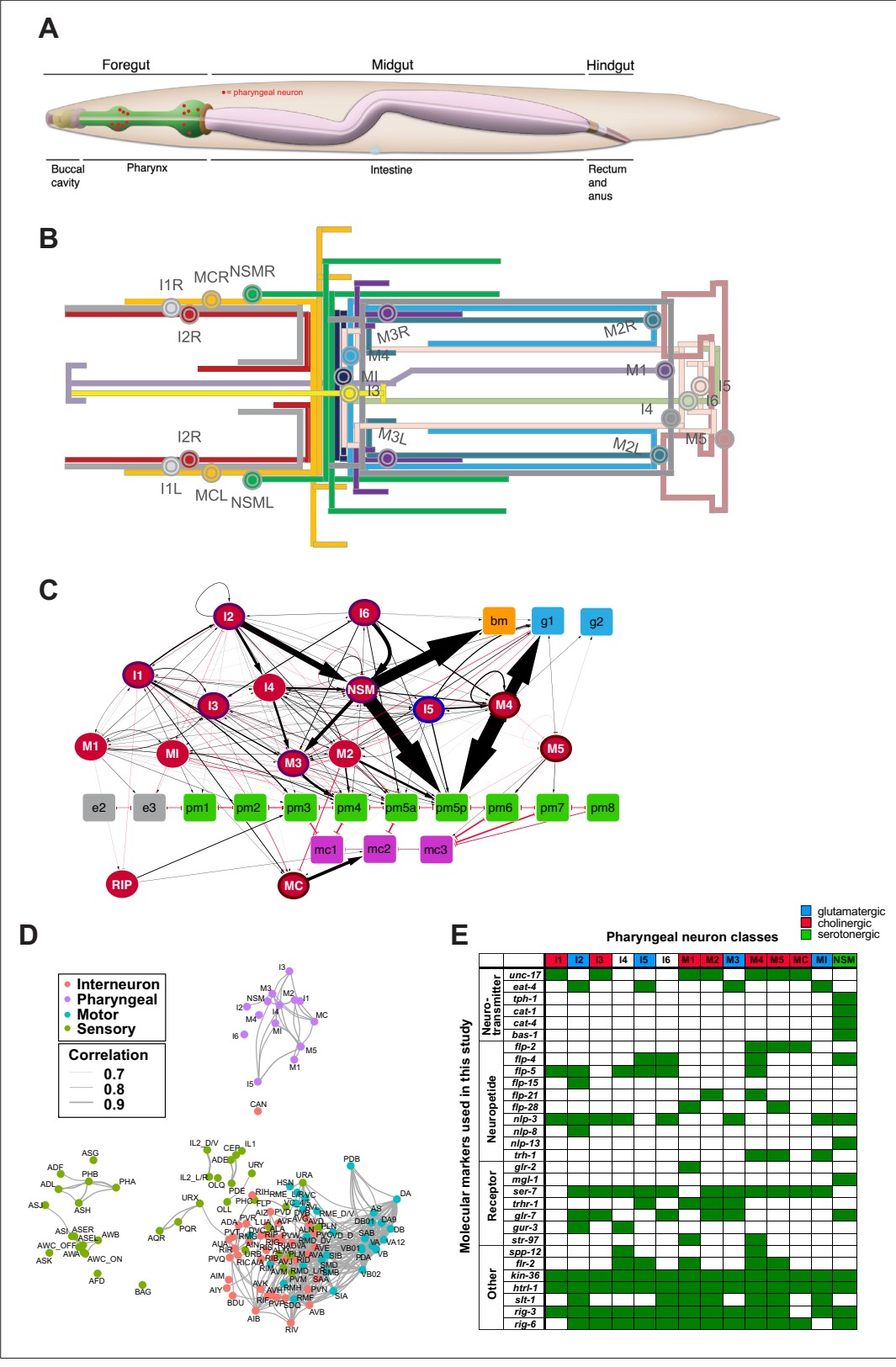

**Figure 1.** The pharyngeal nervous system of *Caenorhabditis elegans*. (**A**) Overview of the *C. elegans* alimentary system from Wormbook (**Hall and Altun, 2007**), with neuronal cell bodies in the pharynx added in red. (**B**) Projection patterns of pharyngeal neurons within the pharynx displayed in the format of a subway map (kindly provided by SJ Cook). (**C**) Full connectome of pharyngeal nervous system, adapted from **Cook et al., 2020**.

*Figure 1 continued on next page*

*Figure 1 continued*

Square nodes are end organs, including muscle (green), marginal cells (fuchsia), gland cells (blue), epithelial cells (gray), and basement membrane (orange). Neurons are red ellipses. Neurons with outlines have either apical (purple), unexposed (brown), or embedded (blue) sensory endings. Directed chemical edges and undirected gap junction edges are represented by black arrows and red lines, respectively. The line width is proportional to the anatomical strength of that connection (# serial sections). The pharyngeal nervous system is connected to the rest of the nervous system through a single neuron pair (RIP). (**D**) Single cell transcriptome similarity between neuron types classes with widths of edges indicating strengths of similarity (Pearson correlation coefficients > 0.7), showing that pharyngeal neurons are more similar to each other than to other neurons in the *C. elegans* nervous system. Reproduced from *Taylor et al., 2021*. (**E**) Molecular markers used in this study for cell fate analysis. See *Supplementary file 1* for information on reporter constructs.

pharyngeal cells by lineage; for example, some muscle and neurons derive from a common mother cell (*Sulston et al., 1983*). The idea of enteric neurons being reflective of an early, primitive state of the nervous system has also been brought forward in the context of comparing enteric nervous systems from widely divergent species (*Furness and Stebbing, 2018*; *Gilbert, 2019*). Specifically, these authors argued that the nerve net-like hydra nervous system displays features of the vertebrate enteric nervous system.

The self-contained and hypothetically primitive state of the *C. elegans* enteric nervous system, with all its unique features, encouraged us to use this system as a model to probe several concepts of neuronal identity specification that have emerged from the centralized, non-pharyngeal nervous system of *C. elegans*: (1) The first is the concept of terminal selectors, transcription factors that act in a master-regulatory manner to coordinately control the many identity features of a terminally differentiating neuron (*Hobert, 2016*). Are members of terminal gene batteries in each pharyngeal neuron also controlled in a coordinated manner, via terminal selectors? One study in the NSM neurons provided some limited evidence in this regard (*Zhang et al., 2014*), but how broadly this applies throughout the pharyngeal nervous system was less clear. (2) Second, homeodomain transcription factors have a predominant role as terminal selectors of neuronal identity in the non-pharyngeal nervous system (*Hobert, 2021*; *Reilly et al., 2020*). A recent cataloguing of the expression patterns of all homeodomain proteins in the *C. elegans* genome revealed that each one of the *C. elegans*' 118 neuron classes, including the pharyngeal neurons, display a unique combinatorial signature of homeodomain protein expression (*Reilly et al., 2020*). Are all pharyngeal neurons indeed specified by homeodomain protein combinations? (3) Lastly, there is evidence from both *C. elegans* (*Berghoff et al., 2021*; *Pereira et al., 2015*) and other systems (*Brunet and Pattyn, 2002*) that synaptically connected neurons are often specified by the same transcription factor, suggesting that such transcription factors may be involved in assembling neurons in functional circuitry. We have termed such transcription factors 'circuit organizers' (*Berghoff et al., 2021*; *Pereira et al., 2015*) and sought to test whether the isolated pharyngeal circuitry is similarly specified by a circuit organizer transcription factor.

In this paper, we show that all three predictions are fulfilled in the context of the nematode's pharyngeal/enteric nervous system. We show that a *C. elegans* ortholog of the Sine oculis/Six1/Six2 homeobox gene, *ceh-34*, is expressed in all pharyngeal neurons from their birth throughout their life time. Its expression is induced by the foregut organ selector gene *pha-4/FoxA*. We demonstrate that *ceh-34* initiates and maintains the terminally differentiated state of all synaptically connected pharyngeal neurons, that *ceh-34* is required to assemble and maintain pharyngeal neuron architecture and that in distinct pharyngeal neuron types, *ceh-34* cooperates with distinct homeobox genes to specify and maintain their respective identity. Taken together, our studies further substantiate overarching themes of nervous system development and potentially provide insights into the evolution of nervous systems.

## Results

### Expression of paralogous genes *ceh-34* and *ceh-33*, the two *C. elegans* Sine oculis/Six1/2 orthologs

Genome sequence mining revealed several *C. elegans* homologs of the Sine oculis/Six family of homeodomain proteins (*Ruvkun and Hobert, 1998*) whose founding member was first identified in *Drosophila* for its role in eye patterning (*Cheyette et al., 1994*). This specific homeodomain transcription factor family is characterized by the presence of a conserved domain, located N-terminally to the DNA binding homeodomain, the ~150 amino acid-long SIX domain, involved in both protein-DNA, as well as protein-protein interactions (*Kumar, 2009*; *Patrick et al., 2013*). *C. elegans* Six-type homeodomain proteins fall into several, phylogenetically conserved families, the Sine oculis/Six1/2, the Six4/5, and the Six3/6 subfamily (*Dozier et al., 2001*; *Kumar, 2009*). We focus here on the Sine oculis subfamily.

Through the analysis of multiple nematode genome sequences, we found that nematodes generally contain a single ortholog of the Sine oculis/Six1/2 subfamily of SIX homeodomain proteins, but that this locus has duplicated in the *Caenorhabditis* genus into two immediately adjacent paralogs, *ceh-33* and *ceh-34* (*Figure 2*, *Figure 2—figure supplement 1*). Using a fosmid-based reporter in which the *ceh-33* locus is tagged with *gfp*, we found that the CEH-33 protein shows no expression in the nervous system within or outside the pharynx at any developmental stage. The only observed expression was in a subset of head muscle cells (*Figure 2B*).

A previously described reporter construct that contains the coding region and 3.8 kb of promoter region of the neighboring *ceh-34* gene showed expression in all pharyngeal neurons (*Hirose et al., 2010*). A fosmid-based reporter shows the same expression pattern (*Reilly et al., 2020*). To further confirm this strikingly selective pattern of neuronal expression and also to examine expression at different developmental stages with the best possible reagent, we used the CRISPR/Cas9 system to engineer a reporter allele of *ceh-34*. This reporter shows the same expression as the fosmid-based reporter in all pharyngeal neurons, but no other neurons (*Figure 2B*). The *ceh-34* reporter is turned on in the embryo at around the time of birth of pharyngeal neurons and is maintained in all pharyngeal neurons throughout all larval and adult stages (*Figure 2B*).

The remarkable restriction of *ceh-34* expression within the nervous system to all pharyngeal neurons prompted us to ask whether the Forkhead transcription factor *pha-4*, an organ selector gene involved in early patterning of the pharynx (*Gaudet and Mango, 2002*; *Horner et al., 1998*; *Kalb et al., 1998*; *Mango et al., 1994*), is required for *ceh-34* expression. Crossing a *ceh-34* reporter into *pha-4(q490)* mutant animals, we indeed observed a loss of *ceh-34* expression (*Figure 2C*). Conversely, *ceh-34* does not affect *pha-4* expression (*Figure 2—figure supplement 2A*). The regulation of *ceh-34* by *pha-4* mirrors the effects that *pha-4* has on the expression of other transcription factors that control terminal differentiation of other tissue types in the pharynx, for example, the *ceh-22/NKX* homeobox gene that specifies pharyngeal muscle differentiation (*Vilimas et al., 2004*), or the *hlh-6* bHLH gene that specifies pharyngeal gland differentiation (*Smit et al., 2008*). We furthermore note that a *pha-4* reporter allele that we generated through CRISPR/Cas9 genome engineering is continuously expressed throughout the entire pharyngeal nervous system, at all postembryonic stages (*Figure 2—figure supplement 2B*), raising the possibility that *pha-4* may not only initiate, but also maintain *ceh-34* expression.

### *ceh-34* controls the expression of diverse neurotransmitter signaling pathways in pharyngeal neurons

To begin to assess the function of *ceh-34* in enteric nervous system differentiation, we used CRISPR/Cas9 engineering to generate a null allele of the *ceh-34* locus, *ot1014*, in which the entire *ceh-34* locus is deleted (*Figure 2A*). Previously described *ceh-34* alleles include a hypomorphic splice site allele, *n4796*, and a small deletion allele, *tm3733* (*Amin et al., 2009*; *Hirose et al., 2010*). In our ensuing mutant analysis, we found *ot1014* to be phenotypically indistinguishable from the *tm3733* deletion allele and we therefore used both alleles interchangeably. Both the *ot1014* and *tm3733* alleles result in a completely penetrant early larval arrest phenotype, as expected from loss of pharynx function and resulting inability to feed.

We first asked whether *ceh-34* is required for the generation of pharyngeal neurons. Using a *pha-4* reporter that labels all pharyngeal cells, we observed no obvious differences in the number

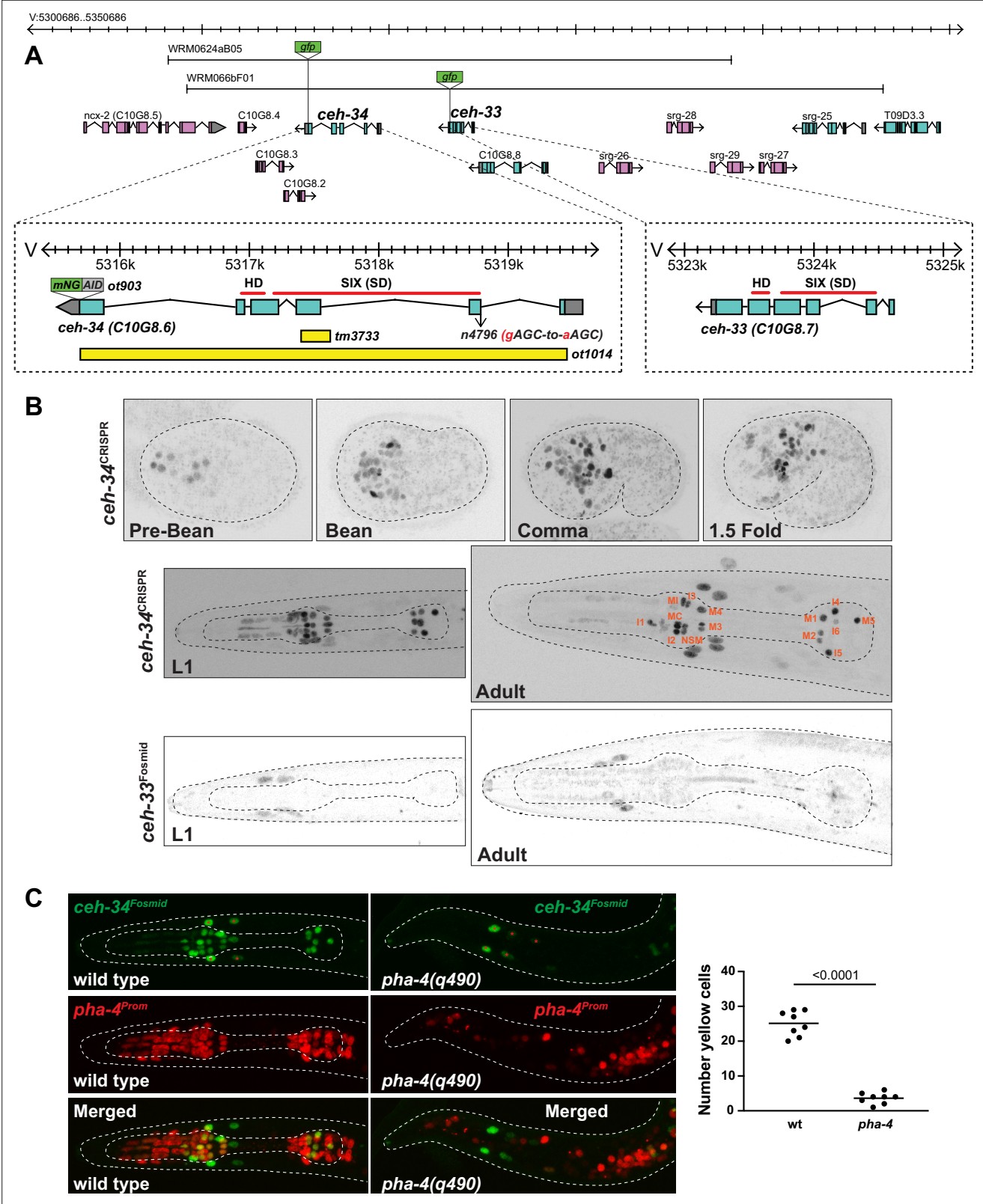

**Figure 2.** *ceh-34* is expressed in all pharyngeal neurons. (**A**) *ceh-33* and *ceh-34* loci showing different alleles and fosmid reporters used in this study. (**B**) Expression of the *ceh-34* CRISPR/Cas-9-engineered reporter allele *ot903* over the course of development. *ceh-33* fosmid reporter (*wgIs575*) shows expression in a subset of head muscle cells. (**C**) Pharynx organ selector *pha-4* controls *ceh-34* expression (as analyzed with the *wgIs524* transgene). Animals were scored at the L1 stage. Presumptive 'pharyngeal cells' in *pha-4* mutant are marked with a red *pha-4* promoter fusion (*stIs10077*). Cells co-

*Figure 2 continued on next page*

*Figure 2 continued*

expressing *ceh-34* and *pha-4* were counted (yellow cells). *ceh-34* expression in head muscle cells, marked with red asterisk, is not affected since they do not express *pha-4*.

The online version of this article includes the following figure supplement(s) for figure 2:

**Figure supplement 1.** Evolution of Sine oculis orthologs in nematodes.

**Figure supplement 2.** Expression of the organ selector *pha-4*.

of pharyngeal cells in *ceh-34* mutants (*Figure 2—figure supplement 2A*). Moreover, the expression of the pan-neuronal genes *ric-4/SNAP25*, *ric-19/ICA1*, *rab-3/RAB3*, and *unc-11/AP180* is unaffected (*Figure 3A*), indicating that pharyngeal neurons are generated and properly execute a generic neuronal differentiation program.

We next assessed the effect of *ceh-34* on a large collection of neuron type-specific molecular identity features (illustrated in *Figure 1E*). To this end, we first focused on the signaling capacities of all pharyngeal neurons. In the vertebrate enteric nervous system, each individual neuron class is distinguished by its unique set of signaling molecules, from classic neurotransmitters to neuropeptides (*Morarach et al., 2021*). The same applies to all neuron classes in the pharyngeal/enteric nervous system of *C. elegans* (*Horvitz et al., 1982*; *Pereira et al., 2015*; *Serrano-Saiz et al., 2013*; *Taylor et al., 2021*). We first considered the three classic neurotransmitters ACh, Glu, and serotonin which are employed in *C. elegans* much like in the enteric nervous system of other species: 7 of the 14 pharyngeal neuron classes use ACh as neurotransmitter, 4 use Glu, and 1 uses serotonin (NSM) (*Horvitz et al., 1982*; *Pereira et al., 2015*; *Serrano-Saiz et al., 2013*). Of those neurotransmitters, serotonin is perhaps the best studied neurotransmitter both in the vertebrate enteric nervous system (*Gershon, 2013*) and in the nematode pharyngeal nervous system (*Horvitz et al., 1982*; *Ishita et al., 2020*; *Song and Avery, 2013*). Acquisition of ACh, Glu, and serotonin neurotransmitter identity features can be visualized through the expression of a number of enzymes and transporters: *unc-17/VAChT* for cholinergic identity, *eat-4/VGluT* for glutamatergic identity, and *tph-1/TPH*, *cat-1/VMAT*, *bas-1/AAAD*, and *cat-4/GCH* for serotonergic identity. We examined the expression of all these markers, using various reporter genes, in all pharyngeal neurons of *ceh-34* mutant animals and found that the seven cholinergic, four glutamatergic, and single serotonergic neuron classes fail to acquire their respective neurotransmitter identity (*Figure 3B*). We schematize these results in the context of a pharyngeal circuit diagram to illustrate the systemic nature of neurotransmission defects in *ceh-34* mutants (*Figure 3C*).

We note that while in most cases, the effect of *ceh-34* on expression of neurotransmitter systems (as well as other identity markers described in ensuing sections) is fully penetrant and fully expressive, in some cases reporter expression is diminished, but not completely eliminated. At the end of this paper, we describe cofactors for *ceh-34* which may be partially able to compensate for loss of *ceh-34* function.

As in vertebrate enteric nervous systems (*Drokhlyansky et al., 2020*), pharyngeal neurons also display highly patterned expression of various neurotransmitter receptors (*Taylor et al., 2021*). We analyzed the *ceh-34*-dependence of three representative receptors, the serotonin receptor *ser-7*, the ortholog of vertebrate HTR7, as well as a metabotropic and an ionotropic Glu receptor, *mgl-1* and *glr-2*. Each of these receptors is expressed in specific subsets of pharyngeal neurons and *ser-7* is known to control feeding behavior (*Hobson et al., 2006*; *Song and Avery, 2012*; *Song and Avery, 2013*). Using a combination of reporter transgenes and CRISPR/Cas9-engineered *gfp* reporter alleles, we found that the expression of *mgl-1*, *glr-2*, and *ser-7* is strongly affected in different pharyngeal neuron types, if not entirely abrogated upon removal of *ceh-34* (*Figure 4A–C*).

## *ceh-34* controls diverse neuropeptidergic identities of pharyngeal neurons

The function of vertebrate enteric nervous systems is modulated by a number of prominent neuropeptidergic signaling systems (*Abot et al., 2018*; *Llewellyn-Smith, 1989*). In some cases, both neuropeptide and receptor are expressed in the vertebrate enteric nervous system, while in others, the peptide is produced elsewhere but acts on neuropeptide receptors located in the enteric nervous system. Likewise, the *C. elegans* pharyngeal nervous system expresses a great diversity of neuropeptides and

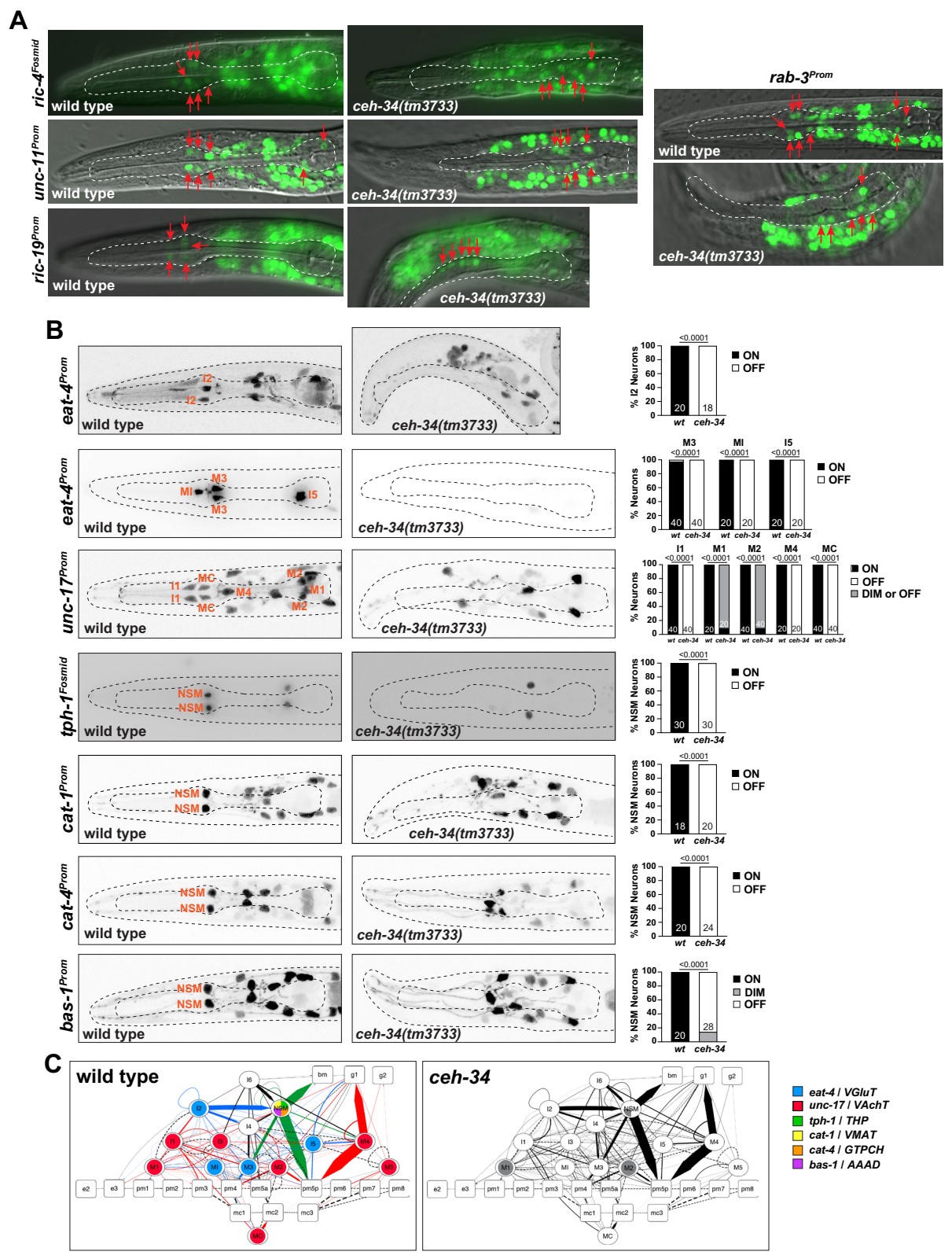

**Figure 3.** Pharyngeal neurons are generated in *ceh-34* mutants but lose their neurotransmitter identity. (**A**) Pictures at the L1 stage showing expression of pan-neuronal reporter transgenes that monitor *ric-4 (otIs350)*, *ric-19 (otIs380)*, *unc-11 (otIs620)*, and *rab-3 (otIs291)* expression. A single focal plane with a subset of pharyngeal neurons marked with red arrows is shown for clarity. (**B**) *ceh-34* affects the expression of neurotransmitter identity genes. Glutamatergic, cholinergic, and serotonergic identity is lost. Reporter transgenes used are *eat-4 (otIs487, otIs558)*, *unc-17 (otIs661)*, *tph-1 (otIs517)*,

*Figure 3 continued on next page*

*Figure 3 continued*

*cat-1 (otIs221), cat-4 (otIs225), and bas-1 (otIs226)*. Animals were scored at the L1 stage. Statistical analysis was performed using Fisher's exact test or chi-square test. N is indicated within each bar and represents number of neurons scored. (**C**) Circuit diagram summarizing the effect of *ceh-34* on neurotransmitter identity. Nodes are colored to illustrate neurotransmitter identity gene expression. Nodes lose coloring when expression is affected in *ceh-34* mutants (gray indicates partial effect). Edges are colored if the source neuron expresses either *eat-4* (glutamatergic), *unc-17* (cholinergic), or *tph-1* (serotonergic). Edges lose coloring when expression of these genes is affected in the source neuron in *ceh-34* mutants (irrespective of whether the effect is partial or total). Note that in this and ensuing circuit diagrams, the existence of gray edges does not indicate whether those edges are generated properly in *ceh-34* mutants. Directed edges (arrows) represent chemical synapses. Undirected edges (dashed lines) represent electrical synapses. The width of the edge is proportional to the weight of the connection (the number of serial section electron micrographs where a synaptic specialization is observed).

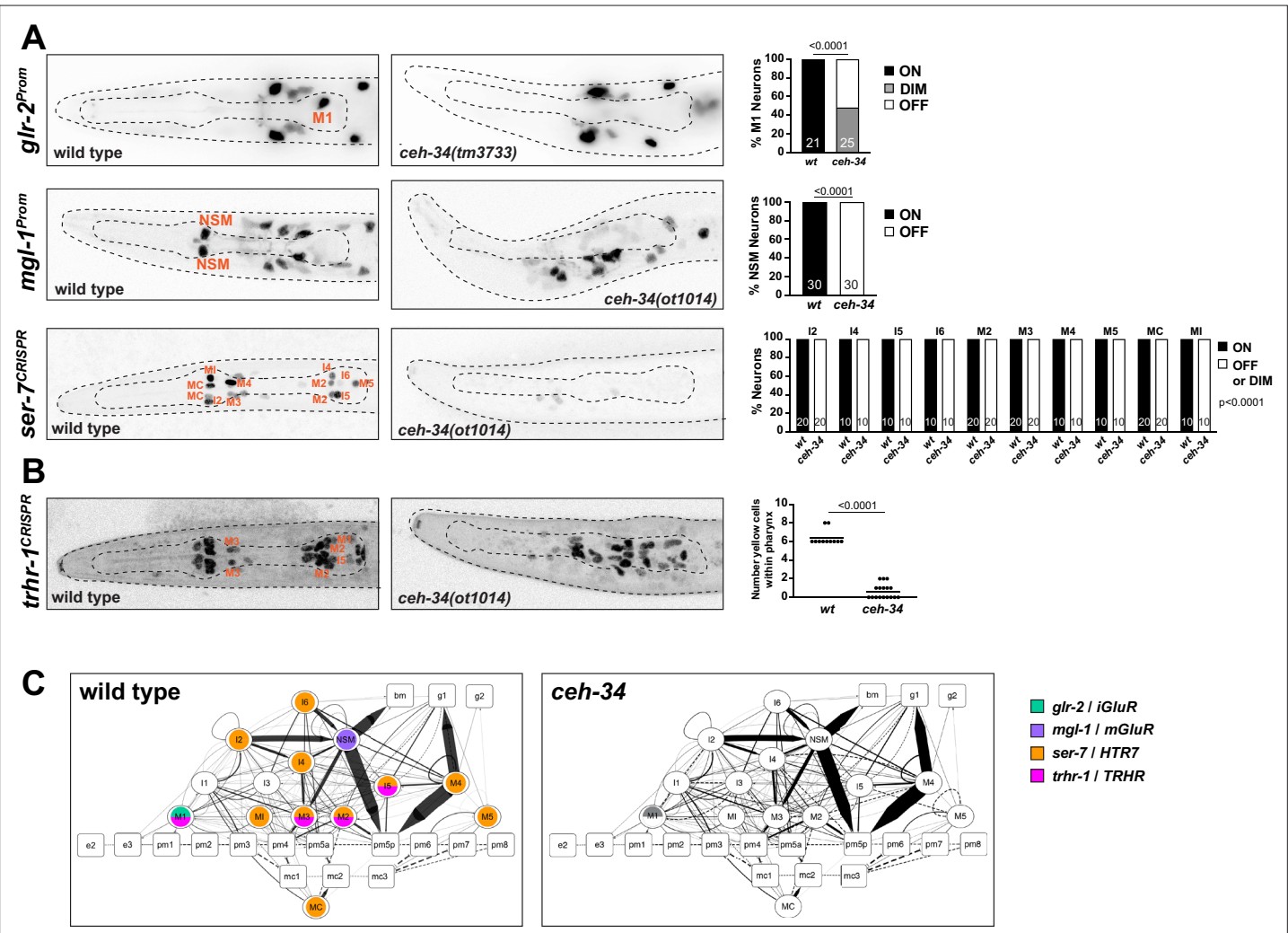

**Figure 4.** *ceh-34* affects the expression of receptors for neurotransmitters and neuropeptides. (**A**) Representative pictures and quantification showing neurotransmitter receptor expression loss in *ceh-34* mutants. Reporter genes used are transgenes *glr-2 (ivIs26), mgl-1 (otIs341)*, and a CRISPR/Cas9-engineered reporter allele for *ser-7 (syb4502)*. Animals were scored at the L1 stage. Statistical analysis was performed using Fisher's exact test or chi-square test. N is indicated within each bar and represents number of neurons scored. (**B**) Representative pictures and quantification showing neuropeptide receptor expression loss in *ceh-34* mutants. Reporter gene used is the CRISPR/Cas9-engineered reporter allele *trhr-1 (syb4453)*. Animals were scored at the L1 stage, with a red pan-neuronal marker (*otIs355*) in the background to facilitate scoring. Number of "yellow" cells (overlap of red pan-neuronal marker and green reporter) within the pharynx were counted. Statistical analysis was performed using unpaired t-test. (**C**) Circuit diagram summarizing the effect of *ceh-34* on neurotransmitter and neuropeptide receptor expression. Nodes lose coloring when expression is affected in *ceh-34* mutants (gray indicates partial effect; *ser-7* and *trhr-1* are colored white in all neurons since identity of neurons with partial effect is not known). See legend to *Figure 3* for more information on circuit diagram features.

neuropeptide receptors, including homologs of neuropeptide signaling systems that function in the vertebrate enteric nervous system (*Taylor et al., 2021*). In fact, each pharyngeal neuron expresses a unique combination of neuropeptides and their receptor proteins (*Taylor et al., 2021*; *Figure 5*).

To test whether *ceh-34* affects the neuron type-specific expression of various neuropeptidergic signaling systems, we first examined the expression of the phylogenetically conservedthyrotropin-releasing hormone (TRH) signaling axis that is important in stimulating gastrointestinal motility in vertebrates (*Abot et al., 2018*). *C. elegans* homologs of either the thyrotropin-releasing hormone (TRH-1) or its receptor (TRHR-1) are expressed in the pharyngeal nervous system (*Hunt-Newbury et al., 2007*; *Van Sinay et al., 2017*), a notion we confirmed and extended with CRISPR/Cas9-engineered reporter alleles, showing that *trh-1* is expressed in the MI, M4, and M5 neurons and *trhr-1* in the M1, M2, M3, and I5 neurons (*Figure 4* and *Figure 5*). We found that the expression of the *trh-1* and *trhr-1* reporter alleles in these pharyngeal neurons is strongly affected in *ceh-34* mutants (*Figure 4* and *Figure 5A*).

In addition to this deeply conserved neuropeptidergic system, we also tested the expression of a cohort of neuropeptides from the FMRFamide family (*flp-2, flp-4, flp-5, flp-15, flp-21, flp-28*) and other miscellaneous neuropeptides (*nlp-3, nlp-8, nlp-13*). The expression of these nine neuropeptides is neuron type-specific, but in aggregate they cover the entire pharyngeal nervous system, often with unique cell type-specific combinations (schematized in *Figure 5B*). We found that the expression of all of these nine neuropeptides, analyzed with either reporter transgenes or CRISPR/Cas9 genome-engineered *gfp* reporter alleles, is affected in *ceh-34* mutants (*Figure 5A*). We schematize these results again in the context of a pharyngeal circuit diagram (*Figure 5B*). Together with our analysis of classic neurotransmitters (ACh, Glu, serotonin) and receptors, we conclude that *ceh-34* is required to endow pharyngeal neurons with their neuron type-specific arsenal of signaling molecules and, hence, that *ceh-34* is a critical specifier of several key aspects of pharyngeal neuron identity and function.

## *ceh-34* is required for sensory receptor expression in the pharyngeal nervous system

We sought to extend our analysis of *ceh-34* mutants by examining the expression of other molecular features of pharyngeal neurons. Like neurons in the vertebrate enteric nervous system, many of the pharyngeal neurons are likely internal sensory neurons that perceive sensory information to modulate peristaltic movements of the alimentary tract (*Cook et al., 2020*). While the sensory apparatus of pharyngeal neurons is not well understood, there are several candidate sensory receptors expressed in pharyngeal neurons. A gustatory receptor family member, *gur-3*, a possible light receptor (*Bhatla and Horvitz, 2015*), is expressed in two pharyngeal neuron classes and its expression is lost in *ceh-34* mutants (*Figure 6A*). Pharyngeal neurons also express the two sole members of the ionotropic sensory receptor family (*Croset et al., 2010*), encoded by *glr-7* and *glr-8* in *C. elegans* (*Brockie et al., 2001*; *Hobert, 2013*). We examined expression of *glr-7* expression, normally observed in six pharyngeal neuron classes (I2, I3, I6, M2, M3, and NSM), in *ceh-34* mutants and found its expression to be completely lost (*Figure 6A*). Finally, we analyzed the expression of *str-97*, a putative chemosensory receptor of the GPCR family which we found to be expressed in several pharyngeal neurons (*Vidal et al., 2018*; *Figure 6A*). We found that *ceh-34* is required for proper *str-97* expression (*Figure 6A*).

## *ceh-34* is required for the expression of antimicrobial defense machinery

One deeply conserved feature of the gut and its associated nervous system is its engagement in antimicrobial defense, either directly through the release of antimicrobial peptides or through the employment of signaling systems that activate the immune system (*Klimovich and Bosch, 2018*; *Muniz et al., 2012*). Similar defense strategies operate in *C. elegans* (*Dierking et al., 2016*). Aside from the epithelial cells of the intestine, the pharyngeal nervous system appears to play a direct role in these microbial control mechanisms, as inferred by the pharyngeal neuron expression of specific proteins implicated in antimicrobial defense. For example, pharyngeal neurons express a hormone, FLR-2, homologous to glycoprotein hormone alpha subunit, that signals to the intestine to orchestrate antimicrobial defense (*Oishi et al., 2009*). Pharyngeal neurons also secrete pore-forming polypeptides that directly kill bacteria, such as the SPP-12 protein (*Hoeckendorf et al., 2012*), as well as a defensin-type antimicrobial peptide, ABF-2 and fungal-induced peptides (FIPR proteins; *Kato et al. 2002*; *Taylor et al., 2021*). Our scRNA transcriptome analysis (*Taylor et al., 2021*) revealed

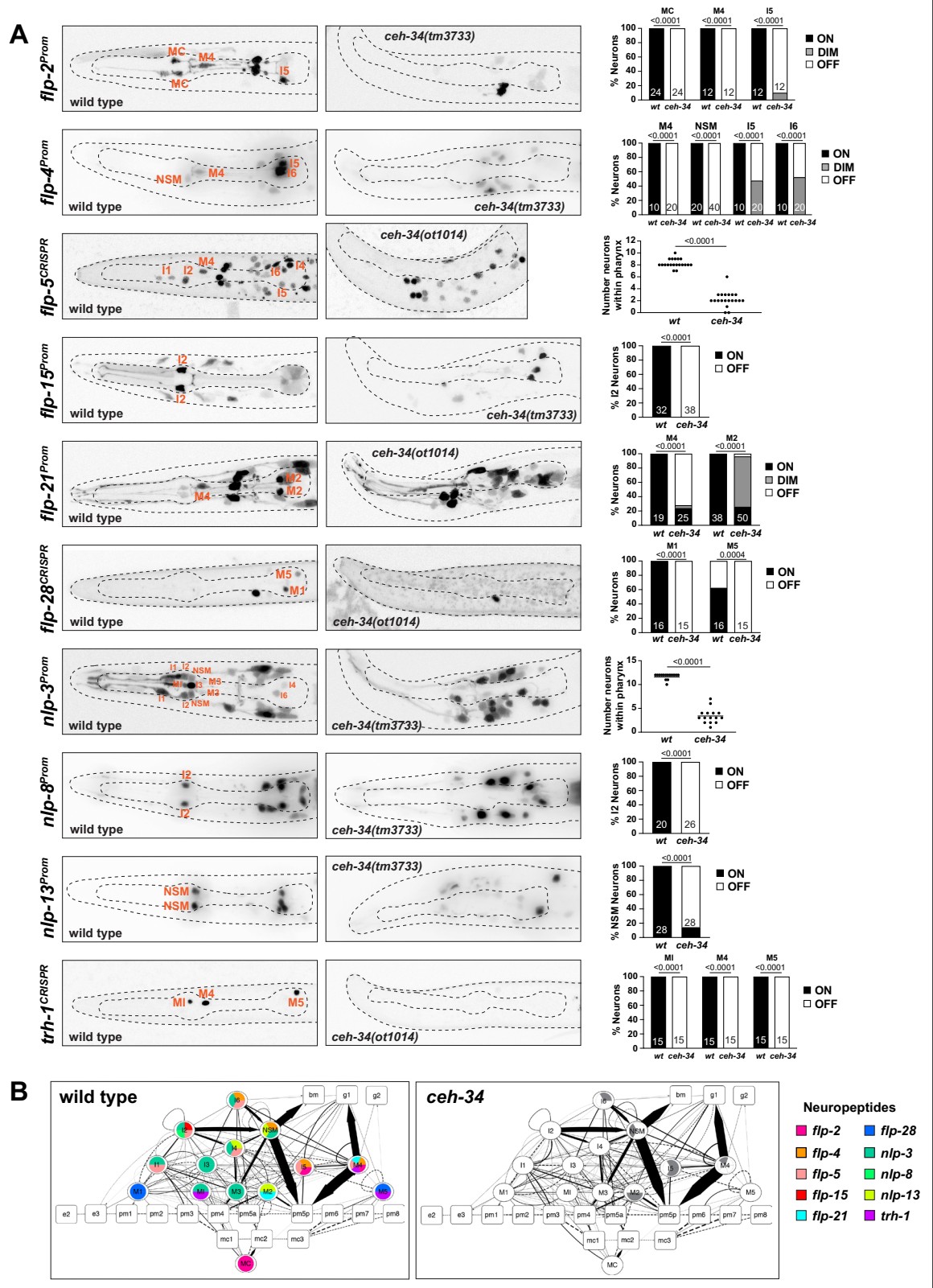

**Figure 5.** *ceh-34* affects neuropeptidergic identity of pharyngeal neurons. (**A**) Representative pictures and quantification showing expression of 10 different neuropeptides is affected in *ceh-34* mutants. Reporter genes used are transgenic reporters for *flp-2 (ynIs57), flp-4 (ynIs30), flp-15 (ynIs45), flp-21 (ynIs80), nlp-3 (otIs695), nlp-8 (otIs711)*, and *nlp-13 (otIs742)* and CRISPR/Cas9-engineered reporter alleles for *flp-5 (syb4513), flp-28 (syb3207)*, and *trh-1 (syb4421)*. Animals were scored at the L1 stage. Statistical analysis was performed using Fisher's exact test, chi-square test, or unpaired t-test. N is

*Figure 5 continued on next page*

*Figure 5 continued*

indicated within each bar and represents number of neurons scored. (**B**) Circuit diagram summarizing the effect of *ceh-34* on neuropeptide expression. Nodes lose coloring when neuropeptide expression is affected in *ceh-34* mutants (gray indicates partial effect; *flp-5* and *nlp-3* are colored white in all neurons since identity of neurons with partial effect is not known). See legend to *Figure 3* for more information on circuit diagram features.

pharyngeal neuron expression of another saposin-related secreted protein, which we named *htrl-1* (see Materials and methods). We visualized expression of these signaling molecules, using a promoter fusion for *spp-12* (*Hoeckendorf et al., 2012*) and CRISPR/Cas9-engineered *gfp* reporter alleles for *flr-2* and *htrl-1* (*Figure 6B*). We found that the *spp-12* reporter is expressed in I4 and M4 neurons, the *flr-2::SL2::gfp::h2b* reporter allele is expressed in five pharyngeal neuron classes (I4, I5, M1, M4, and M5), and the *htrl-1::SL2::gfp::h2b* reporter allele is expressed in all pharyngeal neurons (and in all other pharyngeal cells, but nowhere outside the pharynx at the L1 stage; *Figure 6B*). A notable feature of the extrapharyngeal expression of the *flr-2* reporter allele is expression in the AVL and DVB neurons (*Figure 6B*), the only extrapharyngeal neurons of the *C. elegans* nervous system that innervate gut tissue (not the foregut, but the midgut; *White et al., 1986*). Crossing these reporters into a *ceh-34* mutant background, we found that expression of all three genes (*spp-12, flr-2, htrl-1*) is severely reduced or eliminated in pharyngeal neurons (*Figure 6B*).

## *ceh-34* is required for the expression of pan-pharyngeal nervous system genes

In addition to investigating the *ceh-34*-dependence of genes that fall into specific functional categories, we also sought to capitalize on the recently released single cell transcriptome profiling of the entire *C. elegans* nervous system that included the entire pharyngeal nervous system (*Taylor et al., 2021*). This analysis had shown that the molecular signatures of pharyngeal neurons are more similar to each other than to other neurons in the nervous system (*Taylor et al., 2021*; *Figure 1D*). This pattern is driven, in part, by a number of previously entirely uncharacterized genes with very broad, if not pan-pharyngeal nervous system expression (but no expression in non-pharyngeal neurons), including the above-mentioned saposin-related *htrl-1* gene, small cell surface proteins (e.g. C54E4.4) and a novel receptor tyrosine kinase, which we named *kin-36* (*Figure 6—figure supplement 1*). To confirm this expression pattern, we tagged the *kin-36* locus with a *gfp::H2B::SL2* cassette at its 5' end, using CRISPR/Cas9 genome engineering. We found that *kin-36* indeed displays pan-pharyngeal neuron expression; expression is also observed in all other pharyngeal cell types, but no cell types outside the pharynx, except some unidentified vulval cells (*Figure 6C*). Crossing the *kin-36* reporter allele into *ceh-34* mutants, we observed what appears to be selective loss of *kin-36* expression from many, albeit not all pharyngeal neurons, a similar effect to what we observed for *htrl-1* (*Figure 6B and C*).

We conclude that *ceh-34* is required for the adoption of a broad palette of individual molecular features of all pharyngeal neurons, consistent with a role as a terminal selector of neuronal identity of all pharyngeal neurons. Like other terminal selectors, *ceh-34* only affects neuron type-specific features, but not features that are expressed by all neurons throughout the nervous system.

## *ceh-34* is continuously required to maintain the differentiated and functional state of enteric neurons

The effect of *ceh-34* on terminal marker expression and its continuous expression throughout the life of all pharyngeal neurons suggests that, like other terminal selectors in the non-pharyngeal nervous system, *ceh-34* may not only initiate but also maintain the terminally differentiated state. To test this possibility, we generated a conditional *ceh-34* allele that allowed us to deplete CEH-34 protein post-developmentally. To this end, we inserted an auxin-inducible degron (AID) (*Zhang et al., 2015*) into the *ceh-34* locus using CRISPR/Cas9 genome engineering. Together with a ubiquitously expressed $_{At}$TIR1$^{F79G}$ ubiquitin ligase that recognizes the degron (*rps-28* driver; *cshIs140*; *Hills-Muckey et al., 2022*), this approach allows for temporal depletion of CEH-34 protein through addition of an auxin derivative (5-Ph-IAA) to the worm diet (*Figure 7A and B*). Postembryonic 5-Ph-IAA addition at either larval or adult stages resulted in downregulation of four tested markers for the differentiated state of different pharyngeal neuron classes, *eat-4/VGluT, unc-17/VAChT, ser-7*, and *spp-12* (*Figure 7B and C*). These effects are not as strong as in null mutants, but this is likely due to incomplete CEH-34 protein

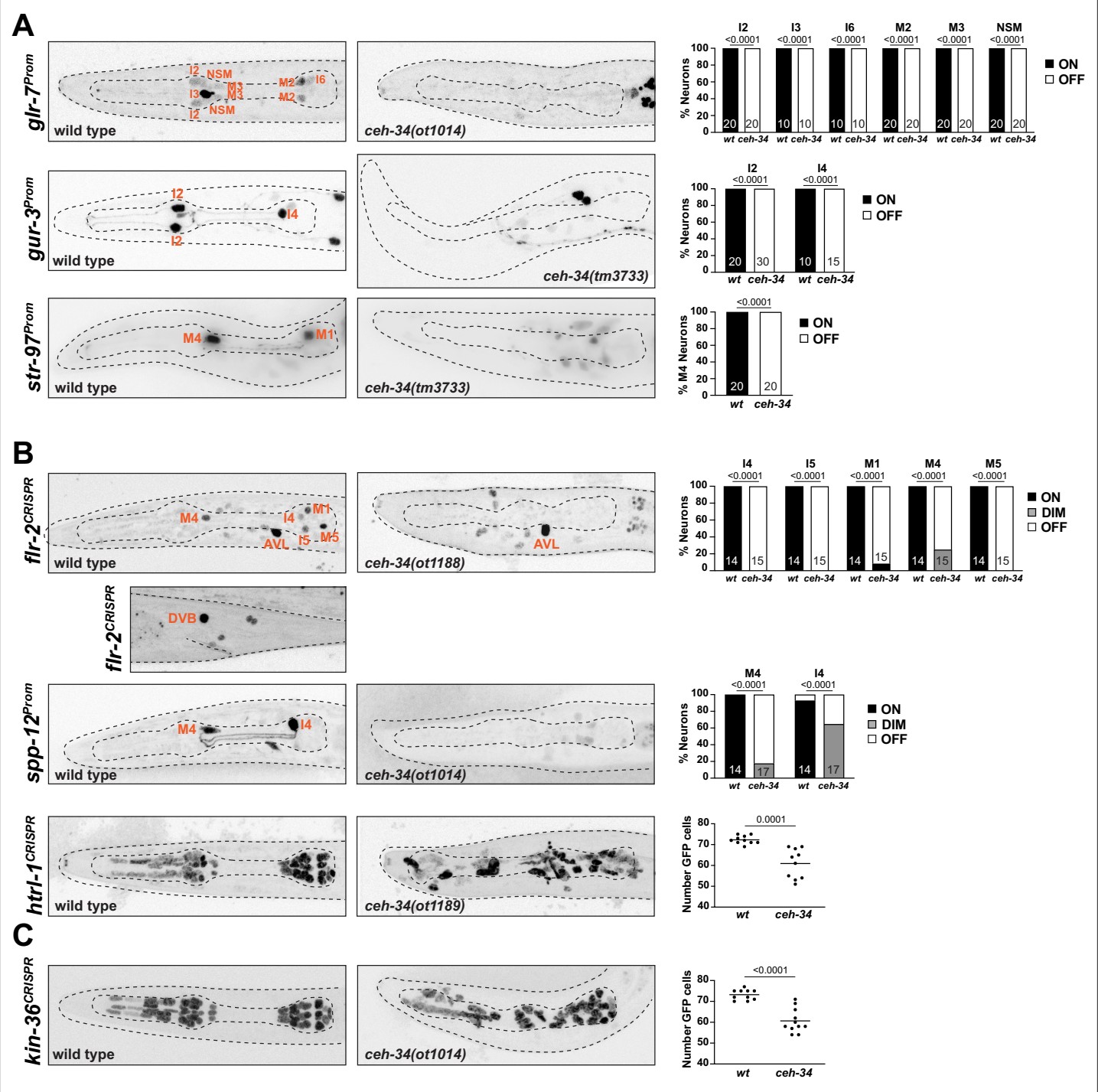

**Figure 6.** *ceh-34* affects other identity features of pharyngeal neurons. (**A**) Representative pictures and quantification showing sensory receptor expression loss in *ceh-34* mutants. Reporter transgenes used are *glr-7 (otIs809), gur-3 (nIs780)*, and *str-97 (otIs716)*. *str-97 (otIs716)* is expressed in M1 in adult animals, but also in M4 in first larval stage animals. Expression in M4 is lost in *ceh-34* mutant, but expression in M1 could not be reliably scored. Animals were scored at the L1 stage. Statistical analysis was performed using Fisher's exact test. N is indicated within each bar and represents number of neurons scored. (**B**) Representative pictures and quantification showing effect of *ceh-34* on antimicrobial defense genes. Reporter genes used are *spp-12 (otIs868)* and CRISPR/Cas9-engineered reporter alleles for *flr-2 (syb4861) and htrl-1 (syb4895)*. Animals were scored at the L1 stage. Statistical analysis was performed using Fisher's exact test, chi-square test, or unpaired t-test. N is indicated within each bar and represents number of neurons scored. (**C**) Representative pictures and quantification showing effect of *ceh-34* on pan-pharyngeal genes. Reporter gene is the CRISPR/Cas9-engineered reporter allele *kin-36 (syb4677)*. Animals were scored at the L1 stage. Statistical analysis was performed using unpaired t-test.

The online version of this article includes the following figure supplement(s) for figure 6:

*Figure 6 continued on next page*

*Figure 6 continued*

**Figure supplement 1.** Summary of previous single cell transcriptomic analysis of genes broadly expressed in pharyngeal neurons.

**Figure supplement 2.** *ceh-34* affects *ceh-28* expression.

depletion, since constitutive 5-Ph-IAA exposure from parental stages throughout all developmental stages also produces only limited defects in expression of these markers.

We also assessed the functional consequences of CEH-34 protein depletion in adult animals, as well as larval stage animals. Using again the AID approach, we found that postembryonic CEH-34 depletion at either larval or adult stages results in substantial defects in pharyngeal pumping, as expected from a disruption of enteric nervous system function (*Figure 7D*). We conclude that *ceh-34* is required to maintain differentiated features of pharyngeal neurons and therefore fulfills another key criterion to classify as a terminal selector of pharyngeal neuron identity.

## Pharyngeal nervous system architecture is severely disorganized in *ceh-34* mutants

We further extended our analysis of *ceh-34* function by analyzing the anatomy of pharyngeal neuron circuitry in *ceh-34* mutants. This analysis is particularly important in light of the observation that, both in *C. elegans* (*Berghoff et al., 2021*; *Pereira et al., 2015*) and in vertebrates (*Brunet and Pattyn, 2002*), several instances have been described in which synaptically interconnected, but otherwise distinct neurons express the same transcription factor. Such observation suggests that these transcription factors may have a role in assembling neurons into functional circuitry. *ceh-34* represents a particularly extreme version of this scenario, because all *ceh-34(+)* neurons are heavily synaptically interconnected and only make a single robust synaptic contact to the rest of the *C. elegans* nervous system (*Figure 1C*; *Cook et al., 2020*). To assess whether *ceh-34* not only specifies terminal molecular properties of pharyngeal neurons, but also organizes overall circuit architecture, we examined pharyngeal nervous system architecture using fluorescent reporter constructs. Such analysis is complicated by the fact that genes that are selectively expressed in *ceh-34(+)* neurons, and hence could serve as drivers for a fluorescent reporter, are turned off in *ceh-34* null mutant animals, thereby preventing an easy visualization of individual axonal tracts or synaptic contacts. However, we found that the *ceh-34* promoter itself is still expressed until the first larval stage in *ceh-34* null mutants, when these animals arrest development. This *ceh-34* promoter transgene (*otIs762*) reveals that although their identity is not properly specified, as described above, pharyngeal neurons retain the capability to grow neuronal projections in *ceh-34* mutants; however, axonal tracts are severely disorganized (*Figure 8A*).

We also expressed the cytoplasmically localized TagRFP reporter together with a synaptically localized, GFP-tagged CLA-1/Clarinet protein (a synaptic active zone marker; *Xuan et al., 2017*) under control of the *ceh-34* promoter. In wild-type animals, this transgene (*otIs785*) reveals (1) the axonal tracts of the pharyngeal nervous system and (2) synaptic structures that are strongly enriched in the pharyngeal nerve ring (*Figure 8B*). In *ceh-34* null mutants, we observed not only a disruption of axonal tract anatomy, but a severe disorganization of presynaptic clusters throughout the entire pharyngeal nervous system (*Figure 8B*).

Using the CLA-1 synaptic marker, we also asked whether CEH-34 is continuously required to maintain synaptic architecture postembryonically (i.e. after the time when synaptic connections initially form). To this end, we again made use of the AID system and removed CEH-34::mNG::AID either throughout larval stages or in the adult stage. In both cases, we found that such depletion resulted in synaptic clusters becoming disorganized, such that ectopic presynaptic clusters form at ectopic locations in the isthmus of the pharynx (*Figure 8—figure supplement 1*).

Lastly, we made use of a mild *ceh-34* hypomorphic allele, *n4796* (*Hirose et al., 2010*). In *ceh-34(n4796)* animals, the expression of many molecular markers for individual pharyngeal neurons are not affected, allowing to visualize their morphology. Focusing on two neuron types, NSM and I5, we found that in both types, specific axons branches fail to form in *ceh-34(n4796)* animals (*Figure 8C*). We conclude that *ceh-34* is required to establish and maintain proper pharyngeal nervous system architecture.

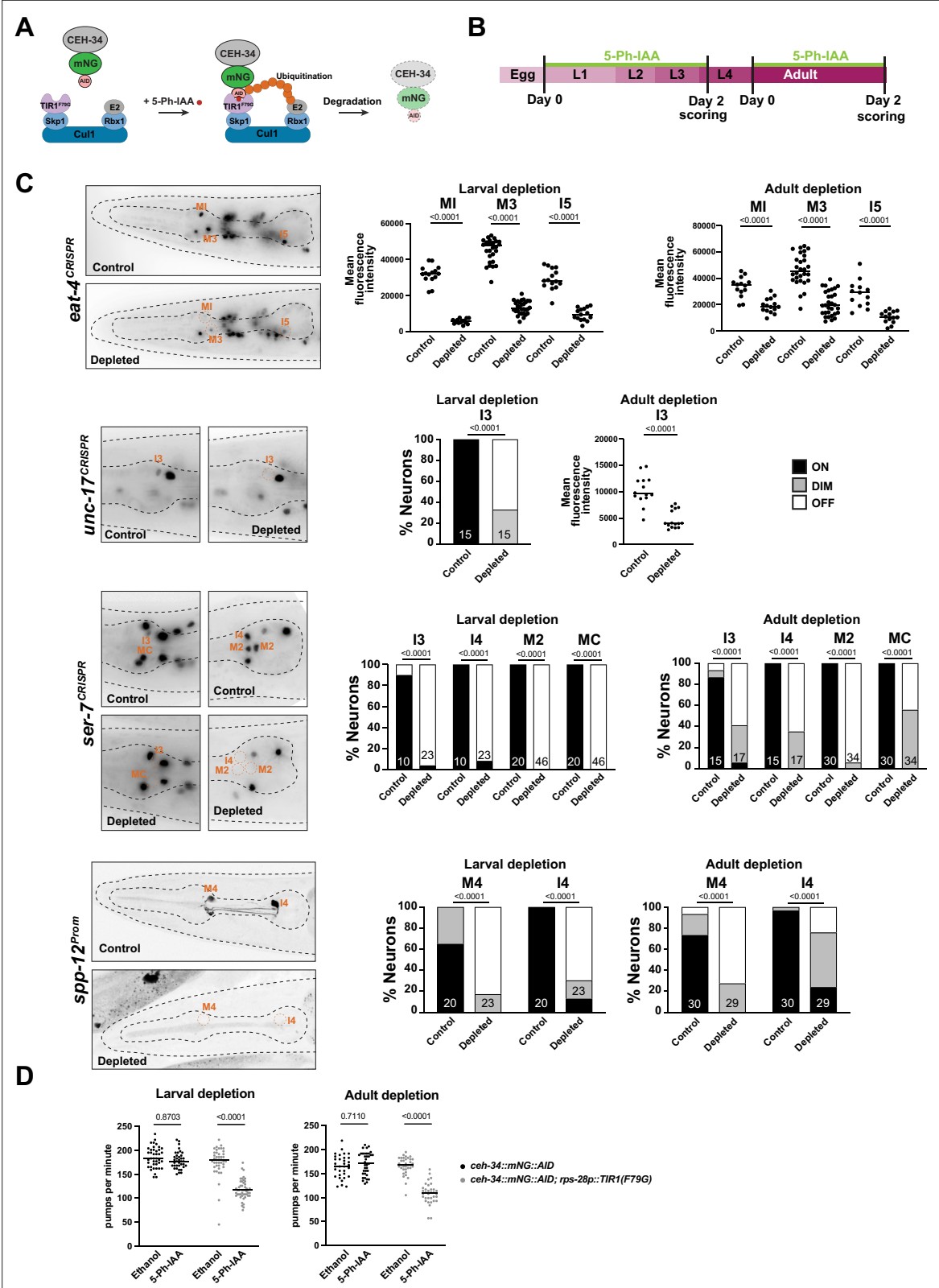

**Figure 7.** *ceh-34* is continuously required to maintain gene expression and function of pharyngeal neurons. (**A**) Schematic of the AIDv2 system (*Hills-Muckey et al., 2022*). Skp1, Cul1, Rbx1, and E2 are phylogenetically conserved components of the E3 ligase complex. TIR1^F79G is a modified plant-specific substrate-recognizing subunit of the E3 ligase complex. In the presence of the auxin analog (5-Ph-IAA), the enzyme TIR1^F79G binds to the AID fused to a protein of interest, leading to ubiquitination and proteasomal degradation of the targeted protein. (**B**) Schematic depicting the 5-Ph-IAA

*Figure 7 continued on next page*

*Figure 7 continued*

treatment. Synchronized populations of worms at the L1 and young adult stage were transferred onto 5-Ph-IAA-coated plates and scored 48 hr later. Worms were expressing TIR1$^{F79G}$ ubiquitously under the *rps-28* promoter (cshIs140). The *ceh-34* locus was tagged with *mNG::AID* (*ot903*). (C) *ceh-34* is required for maintained expression of identity genes. Reporter genes used are *spp-12* (*otIs868*) and CRISPR/Cas-9-engineered reporter alleles for *eat-4* (*syb4257*), *unc-17* (*syb4491*), and *ser-7* (*syb4502*). Since *ceh-34::mNG::AID* (*ot903*) and reporter genes scored are all fluorescent green, and this was obscuring the scoring on ethanol conditions, the control conditions are reporter genes on their own treated with 5-Ph-IAA. Representative pictures of larval depletion are shown on the left and quantification for larval and adult depletion is shown on the right. Quantification is only shown for neurons that were affected. Neurons unaffected by temporally controlled 5-Ph-IAA addition are also unaffected by constitute 5-Ph-IAA addition, indicating an inability to completely deplete CEH-34 protein. Statistical analysis was performed using unpaired t-test, Fisher's exact test, or chi-square test. N is indicated within each bar and represents number of neurons scored. (D) *ceh-34* is required for maintained pharyngeal function. Larval and adult *ceh-34* depletion results in decreased pharyngeal pumping. Statistical analysis was performed using two-way ANOVA.

## *ceh-34* affects the expression of molecules involved in proper wiring of the pharyngeal nervous system

To further explore how *ceh-34* may affect pharyngeal nervous system architecture, we considered the expression of molecules with potential or explicitly demonstrated roles in axon guidance, fasciculation, and/or synapse formation. A genetic analysis of axon guidance and circuit formation has only been conducted in a small number of pharyngeal neurons, mainly the M2 and NSM neuron classes (*Pilon, 2008*). In both neuron classes, the SLT-1 axon guidance cue, the *C. elegans* ortholog of Slit, has been found to be required for proper axon guidance (*Axäng et al., 2008*; *Rauthan et al., 2007*). We extended this phenotypic characterization, finding that other pharyngeal neurons also display axon pathfinding defects in *slt-1* mutants (*Figure 8—figure supplement 2*). To examine potential links between *slt-1* and *ceh-34*, we made use of a promoter::gfp fusion that captures the entire upstream intergenic region of the *slt-1* locus (*Hao et al., 2001*) and which shows selective expression in seven pharyngeal neuron classes (I2, I6, M1, M2, M4, M5, and MI) at the first larval stage. We found that *slt-1* expression is strongly affected in *ceh-34* mutants (*Figure 8D*).

Effects of *ceh-34* on molecules potentially involved in circuit formation are not restricted to *slt-1*. Two Ig superfamily members, *rig-3* and *rig-6* (the sole *C. elegans* ortholog of contactin), have previously been implicated in axon outgrowth and synapse function in the *C. elegans* nervous system (*Babu et al., 2011*; *Bhardwaj et al., 2020*; *Katidou et al., 2013*; *Kim and Emmons, 2017*) and promoter fusion transgenes have indicated their expression in pharyngeal neurons (*Schwarz et al., 2009*). We used CRISPR/Cas9 to tag both loci with *gfp* and found that both genes are broadly expressed in many pharyngeal neurons (*Figures 1E and 8E*). *ceh-34* affects the pharyngeal neuron expression of both *rig-3* and *rig-6* reporter alleles (*Figure 8E*).

## The Six homeodomain cofactor, Eyes absent/Eya, shows limited cooperation with *ceh-34*

To gain further insights into how CEH-34 patterns the identity of a wide array of distinct pharyngeal neuron types, we considered the involvement of cell type-specific cofactors. As a first step, we considered the EYes Absent/EYA protein, a phylogenetically conserved transcriptional co-activator of specific subsets of Six homeodomain proteins (*Ohto et al., 1999*; *Patrick et al., 2013*; *Tadjuidje and Hegde, 2013*). The *C. elegans* ortholog of Eyes absent, called *eya-1* (*Furuya et al., 2005*), also directly physically interacts with CEH-34 protein (*Amin et al., 2009*; *Hirose et al., 2010*). We first examined *eya-1* expression in pharyngeal neurons using a genomic fragment that contains the entire, *gfp*-tagged *eya-1* locus (*Furuya et al., 2005*). We observed expression in all pharyngeal neurons throughout all larval and adult stages, a phenocopy of the *ceh-34* expression pattern (*Figure 9A*), that is also corroborated by our recent scRNA analysis (*Figure 6—figure supplement 1*; *Taylor et al., 2021*). Moreover, we found that *eya-1* expression requires *ceh-34* function, suggesting that *ceh-34* acts in a feedforward configuration to induce its own transcriptional cofactor (*Figure 9B*).

We analyzed the function of *eya-1* in the context of pharyngeal neuron specification. Animals that carry a deletion of a part of the *eya-1* locus, *ok654*, display pharyngeal neuron specification defects, albeit much milder than those observed in *ceh-34* null mutant animals (*Figure 9C*). The larval arrest phenotype of *ceh-34* null mutants is also more penetrant than that of *eya-1* mutants, which are very slow growing, but still homozygous viable (*Furuya et al., 2005*). To exclude the possibility that the *ok654* allele is not a null allele, we used CRISPR/Cas9 to generate a null allele in which the entire locus

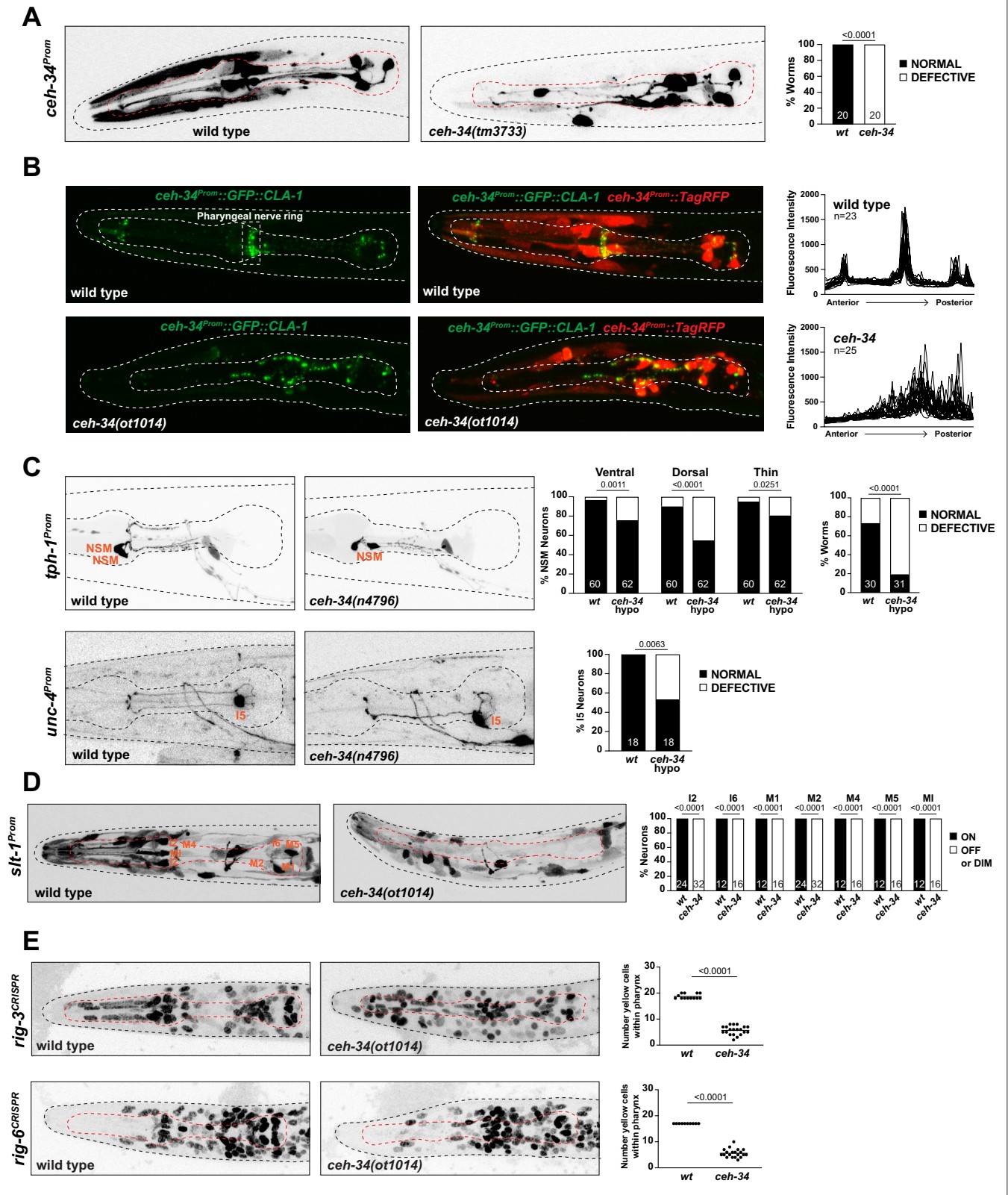

**Figure 8.** *ceh-34* affects the assembly of pharyngeal circuitry. (**A**) *ceh-34* null mutants display disorganized axodendritic projections. Axodendritic projections were scored as a whole rather than by individual neuron because with all the pharyngeal neurons being labeled it was difficult to assign specific projections to individual neurons. Projections were classified as defective only when obviously deviating from the wild-type path. Representative pictures and quantification are shown. Reporter gene is *ceh-34* (*otIs762*). Animals were scored at the L1 stage. Statistical analysis was performed using

*Figure 8 continued on next page*

*Figure 8 continued*

Fisher's exact test. N is indicated within each bar and represents number of worms. (**B**) *ceh-34* null mutants show disorganized pharyngeal nerve ring presynaptic specializations as visualized with CLA-1 puncta. Representative pictures are shown. Quantification (right panels) shows GFP fluorescent intensity profiles along the anterior posterior axis. Reporter gene is *otIs785*. Animals were scored at the L1 stage. (**C**) *ceh-34(n4796)* hypomorph mutants show axonal defects in NSM (top panel) and I5 (bottom panel). Representative pictures and quantification are shown. For NSM, the ventral, dorsal, and thin projection (not visible in picture) were scored separately (graph on the left) and then data was pulled together to indicate the percentage of worms showing any defect (graph on the right). Reporter genes used are *tph-1* (*zdIs13*) and *unc-4* (*otEx7503*). Animals were scored at the L4 stage. Statistical analysis was performed using Fisher's exact test. N is indicated within each bar and represents number of neurons or number of worms. (**D**) *ceh-34* affects expression of the axon guidance cue *slt-1* (*kyIs174*). Representative pictures and quantification are shown. Animals were scored at the L1 stage. Statistical analysis was performed using Fisher's exact test. N is indicated within each bar and represents number of neurons scored. (**E**) *ceh-34* affects expression of CRISPR/Cas9-engineered *gfp* reporter alleles of *rig-3* (*syb4763*) and *rig-6* (*syb4729*), two Ig superfamily members. *rig-3* and *rig-6* are expressed in almost all pharyngeal neurons plus many other cells within and outside the pharynx. Worms were scored with a red pan-neuronal marker (*otIs355*) or a red *ceh-34promoter* fusion (*stIs10447*) in the background to facilitate scoring. Number of yellow cells were counted within the pharynx. Animals were scored at the L1 stage. Statistical analysis was performed using unpaired t-test.

The online version of this article includes the following figure supplement(s) for figure 8:

**Figure supplement 1.** *ceh-34* is required to maintain synapse organization in the pharyngeal nervous system.

**Figure supplement 2.** Axonal defects in *slt-1* mutants.

is deleted. Animals carrying this deletion allele (*ot1197*) display phenotypes that are indistinguishable from those of *ok654* animals. They are still homozygous viable, albeit as slowly growing as *ok654* animals, and they display very similar, limited neuronal cell fate marker defects (*Figure 9C*). Given the milder spectrum of *eya-1* defects compared to *ceh-34* null mutants, we conclude that *ceh-34* may be able to partly function without *eya-1*.

In other organisms, Dachshund proteins are components of Sine oculis/Eya complexes in several cellular contexts (*Hanson, 2001*). However, the sole *C. elegans* ortholog of Dachshund is not expressed in pharyngeal neurons (*Colosimo et al., 2004*; *Taylor et al., 2021*) and *dac-1* null mutants also do not display the larval growth/arrest phenotype characteristic of *ceh-34* and *eya-1* mutants (*Colosimo et al., 2004*). While these observation do not entirely rule out a function for DAC-1 in pharyngeal neurons, it appears unlikely that DAC-1 is an essential cofactor of CEH-34.

## *ceh-34* cooperates with a multitude of other homeobox genes to specify distinct pharyngeal neuron types

How does *ceh-34* activate distinct genes in different pharyngeal neuron types? One obvious possibility is that *ceh-34* cooperates with neuron type-specific cofactors in neuron type-specific terminal selector complexes to drive specific fates. As candidates for such cofactors, we considered homeobox genes, for two reasons: (1) like any other neuron in the *C. elegans* nervous system, each individual pharyngeal neuron expresses a unique combination of homeobox genes, in addition to pan-pharyngeal *ceh-34* (*Reilly et al., 2020*); (2) previous studies had already implicated a few homeobox genes in controlling some select functional or molecular aspects of individual pharyngeal neurons (*Aspöck et al., 2003*; *Feng and Hope, 2013*; *Mörck et al., 2004*; *Ramakrishnan and Okkema, 2014*; *Ray et al., 2008*; *Zhang et al., 2014*). For example, the homeobox gene *ceh-2*, the *C. elegans* ortholog of vertebrate EMX and *Drosophila* Ems, is required for proper function of the M3 neuron, but effects of *ceh-2* on molecular aspects of M3 neuron differentiation had not been reported (*Aspöck et al., 2003*). We therefore set out to analyze homeobox gene function throughout the pharyngeal nervous system and to examine potential interactions of *ceh-34* with other homeobox genes.

### NSM neurons

To ask whether distinct pharyngeal neuron type-specific homeobox genes cooperate with *ceh-34* in distinct neuronal cell types throughout the pharyngeal nervous system, we made use of a hypomorphic *ceh-34* allele, *n4796* (*Hirose et al., 2010*). Unlike the *ceh-34* null allele, which results in very strong expression defects of all NSM molecular markers (*Figures 3–6*), the *n4796* allele displays only subtle if any marker expression effects on its own (*Figure 10A*). However, when combined with a mutant allele of *unc-86*, a POU homeobox gene that affects many but not all NSM marker genes (*Zhang et al., 2014*), strong synergistic differentiation defects of NSM are observed (*Figure 10A*).

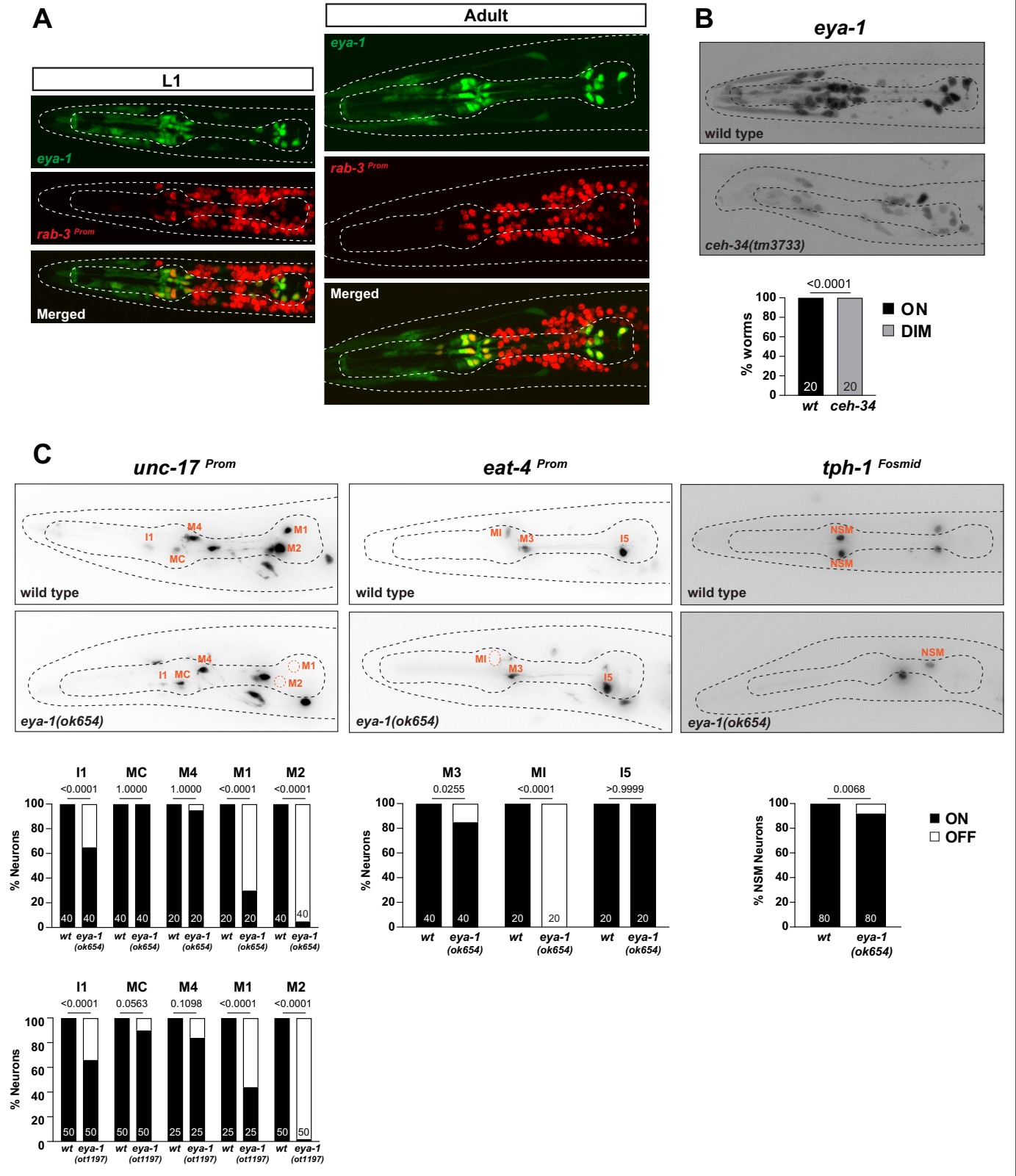

**Figure 9.** Limited involvement of *eya-1* in pharyngeal neuron identity specification. (**A**) *eya-1* is expressed in all pharyngeal neurons throughout the life of the worm. Images of L1 and adult worms showing co-localization of *eya-1* expression (*nIs352*) with the pan-neuronal gene *rab-3* (*otIs355*) in pharyngeal neurons. (**B**) *eya-1* expression is regulated by *ceh-34*. Representative pictures and quantification are shown. Reporter gene is *eya-1* (*nIs352*). Animals were scored at the L1 stage. Statistical analysis was performed using Fisher's exact test. N is indicated within each bar and represents number of

*Figure 9 continued on next page*

*Figure 9 continued*

worms scored. (**C**) *eya-1* mutant animals show defects in neurotransmitter identity specification. Representative images and quantification are shown for *unc-17 (otIs661)*, *eat-4 (otIs487)*, and *tph-1 (otIs517)*. Animals were scored at the L4 stage. Statistical analysis was performed using Fisher's exact test. N is indicated within each bar and represents number of neurons scored.

This genetic interaction mirrors the synergistic effects of *unc-86* and the LIM homeobox gene *ttx-3*, another regulator of NSM differentiation (**Zhang et al., 2014**).

### I1 neurons

A similar genetic interaction between *ceh-34* and *unc-86* is observed in the cholinergic I1 neuron pair, the only other pharyngeal neuron class that also co-expresses *ceh-34* and *unc-86* (**Baumeister et al., 1996**; **Serrano-Saiz et al., 2018**), but which does not express *ttx-3*. While cholinergic identity, visualized via *unc-17/VAChT* expression, is not affected in *unc-86(n846)* single mutants, a combination of the *ceh-34(n4796)* hypomorphic allele with the *unc-86(n846)* mutation results in I1 losing its cholinergic identity (**Figure 10B**).

### I2 neurons

Synergistic interactions are also observed in the glutamatergic I2 neuron pair. Like the I1 neurons, no identity regulators were previously known for this neuron class. Our homeobox gene expression atlas (**Reilly et al., 2020**) showed that the I2 neuron expresses the LIM homeobox gene *ceh-14*, the *C. elegans* ortholog of vertebrate Lhx3/4. No other pharyngeal neuron expresses *ceh-14*. While *ceh-14* single null mutants show no effect on *eat-4/VGluT* expression (the marker of glutamatergic identity), in combination with the *ceh-34(n4796)* hypomorphic allele, a strong synergistic effect on glutamatergic identity acquisition is observed in I2 (**Figure 10C**). Another molecular marker for I2 identity, the neuropeptide *nlp-8*, is also synergistically regulated by *ceh-34* and *ceh-14* (**Figure 10C**).

### I3 neuron

The previously unstudied cholinergic I3 neuron class expresses, in addition to *ceh-34*, the *C. elegans* ortholog of Empty spiracles/EMX, *ceh-2* (**Aspöck et al., 2003**; **Reilly et al., 2020**), as well as the Prospero ortholog *pros-1*, whose function in the nervous system has not previously been examined. We find that in *pros-1* null mutant animals, cholinergic identity of I3, measured with an *unc-17/VAChT* reporter transgene, is not properly acquired (**Figure 10D**). The *ceh-2(ch4)* null allele alone also shows a reduction of *unc-17/VAChT* expression (**Figure 10F**), as well as a reduction of the serotonin receptor *ser-7* expression (**Figure 10E**). In combination with the *ceh-34(n4796)* hypomorphic allele, *unc-17/VAChT* expression, and, hence, cholinergic identity of I3, is eliminated in *ceh-2* mutant animals (**Figure 10F**).

### M3 neuron

Apart from expression in I3, the EMX ortholog *ceh-2* is also expressed in the glutamatergic M3 neurons and is required for proper M3 function (**Aspöck et al., 2003**), but molecular correlates for this functional defect have not previously been identified. We found that in *ceh-2* single mutants, expression of the *eat-4/VGluT* identity marker is affected in M3 (**Figure 11A**).

### M4 neuron

In the cholinergic M4 neuron, the *ceh-28* and *zag-1* homeobox genes have previously been shown to each regulate subsets of M4 identity features (**Ramakrishnan and Okkema, 2014**). Both *zag-1* and *ceh-28* affected *flp-2* expression, but only *zag-1*, and not *ceh-28*, was found to affect *ser-7* expression (**Ramakrishnan and Okkema, 2014**). In contrast, *ceh-28* but not *zag-1* affected *flp-5* expression and neither *zag-1* nor *ceh-28* affected *unc-17* or *flp-21* expression (**Ramakrishnan and Okkema, 2014**). *ceh-34* null mutants show effects on the expression of all the tested *ceh-28* or *zag-1*-dependent (*ser-7*, *flp-2*, and *flp-5*) or -independent markers (*unc-17*, *flp-21*; **Figures 3–5**), indicating that *ceh-34* may collaborate with these homeobox genes to control distinct subsets of M4 differentiation markers. *ceh-34* also affects *ceh-28* expression (**Figure 6—figure supplement 2**).

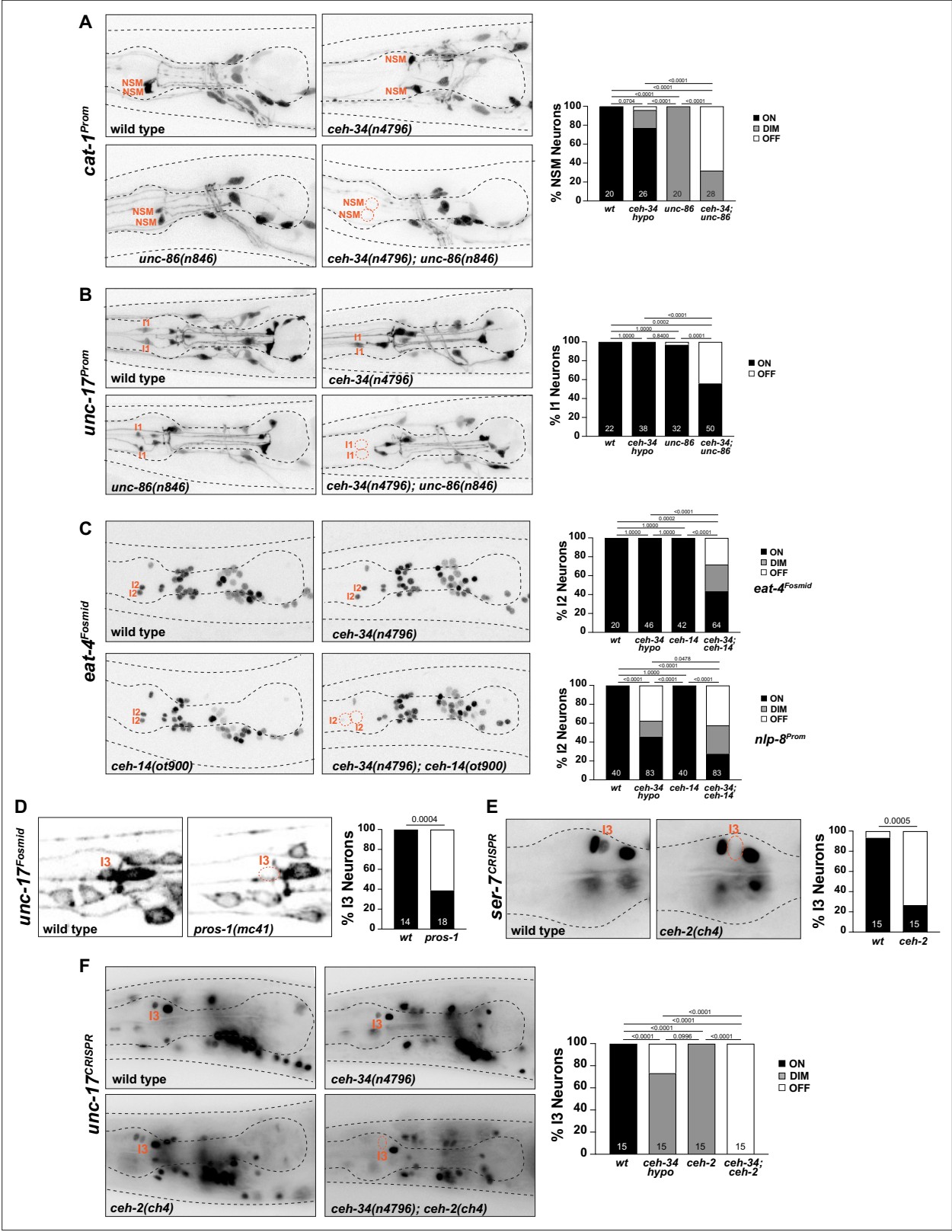

**Figure 10.** *ceh-34* cooperates with homeobox genes to specify distinct pharyngeal neuron types. (**A**) *unc-86* and *ceh-34* synergistically affect NSM differentiation. Representative images and quantification are shown. Reporter gene used is *cat-1* (*otIs224*). Animals were scored at the L4 stage. Statistical analysis was performed using Fisher's exact test. p-values were adjusted with the Holm-Sidak correction for multiple comparisons. N is indicated within each bar and represents number of neurons scored. (**B**) *unc-86* and *ceh-34* show synergistic defects in I1 neuron differentiation.

*Figure 10 continued on next page*

*Figure 10 continued*

Representative images and quantification are shown. Reporter gene used is *unc-17* (*otIs661*). Animals were scored at the L4 stage. Statistical analysis was performed using Fisher's exact test. p-values were adjusted with the Holm-Sidak correction for multiple comparisons. N is indicated within each bar and represents number of neurons scored. (**C**) *ceh-14* and *ceh-34* show synergistic effects on I2 neuron differentiation. Representative images and quantification are shown for *eat-4* (*otIs518*). Bottom graph shows quantification for *nlp-8* (*otIs711*). Animals were scored at the L4 stage. Statistical analysis was performed using Fisher's exact test or chi-square test. p-values were adjusted with the Holm-Sidak correction for multiple comparisons. N is indicated within each bar and represents number of neurons scored. (**D**) *pros-1* affects I3 neuron differentiation. Representative images and quantification are shown. Reporter gene is *unc-17* (*otIs576*). Animals were scored at the L1 stage. Statistical analysis was performed using Fisher's exact test. N is indicated within each bar and represents number of neurons scored. (**E**) *ceh-2* affects I3 neuron differentiation. Representative images and quantification are shown. Reporter gene used is a CRISPR/Cas9-engineered allele for *ser-7* (*syb4502*). Animals were scored at the L4 stage. Statistical analysis was performed using Fisher's exact test. N is indicated within each bar and represents number of neurons scored. (**F**) *ceh-2* and *ceh-34* show synergistic defects in I3 neuron differentiation. Representative images and quantification are shown. Reporter gene used is CRISPR/Cas9-engineered allele for *unc-17* (*syb4491*). Animals were scored at the L4 stage. Statistical analysis was performed using Fisher's exact test. p-values were adjusted with the Holm-Sidak correction for multiple comparisons. N is indicated within each bar and represents number of neurons scored.

## M5 neuron

The cholinergic M5 neuron expresses, in addition to *ceh-34*, the Msh/Msx ortholog *vab-15* (***Reilly et al., 2020***). Since only a hypomorphic allele of *vab-15* was previously available (***Du and Chalfie, 2001***), we generated a molecular null allele, *ot1136*, through CRISPR/Cas9 genome engineering (***Figure 11—figure supplement 1A***). In contrast to the hypomorphic allele, these null mutant animals are very slow growing, but still homozygous viable. We found a complete loss of expression of the cholinergic marker *unc-17/VAChT*, as well as the neuropeptidergic marker, *trh-1*, in *vab-15(ot1136)* animals (***Figure 11B***). Complete loss of marker expression is not an indicator of failure of this neuron to be generated, since crossing a *ceh-34* marker into *vab-15* null mutants revealed the presence and normal *ceh-34* expression of M5 (***Figure 11B***).

## MI neuron

In our previous genome-wide analysis of homeobox gene expression (***Reilly et al., 2020***), we had shown that the glutamatergic MI neuron expresses the sole worm ortholog of the Goosecoid homeobox gene, *ceh-45*. Embryonically, *ceh-45* is expressed in multiple pharyngeal tissues (***Ma et al., 2021***), but its expression resolves to exclusive expression in the MI and I1 neurons (***Reilly et al., 2020***). *ceh-45* had not previously been functionally characterized. We examined *ceh-45* function by generating a null allele using CRISPR/Cas9 genome engineering, *ot1065* (***Figure 11—figure supplement 1A***). *ceh-45* null mutant animals display partially penetrant embryonic lethality, with escapers being slow growing. Mirroring the *ceh-34* defects, we found that glutamatergic identity specification of MI (as assessed by *eat-4/VGluT* expression) is strongly affected in surviving *ceh-45* null mutant animals (***Figure 11C***). Similarly, expression of the neuropeptide *trh-1* is also affected in MI (***Figure 11C***).

In our search for potential *ceh-34* cofactors, we also considered three divergent, non-conserved homeobox genes, *ceh-7*, *ceh-53*, and *ceh-79* that display expression in subsets of pharyngeal neurons (***Reilly et al., 2020***). We generated null alleles for these three genes using CRISPR/Cas9 genome engineering (***Figure 11—figure supplement 1A***). However, we observed no *eat-4/VGluT* or *unc-17/VAChT* expression defects in either of these mutant strains, either alone or in combination with the *ceh-34* hypomorphic allele *n4796* (***Figure 11—figure supplement 1B***).

In conclusion, nine phylogenetically conserved homeobox genes appear to collaborate with *ceh-34* in 8 of the 14 pharyngeal neuron classes to specify their proper identity (summarized in ***Figure 12***). Since the remaining six classes also express specific combinations of homeobox genes (***Reilly et al., 2020***), we anticipate that future analysis will likely reveal homeobox codes throughout the entire pharyngeal nervous system.

## Discussion

We have identified here common, overarching themes in the differentiation of the pharyngeal nervous system, the enteric nervous system of the nematode *C. elegans*. A number of previous studies have identified transcription factors involved in regulating specific differentiation aspects of a small subset of

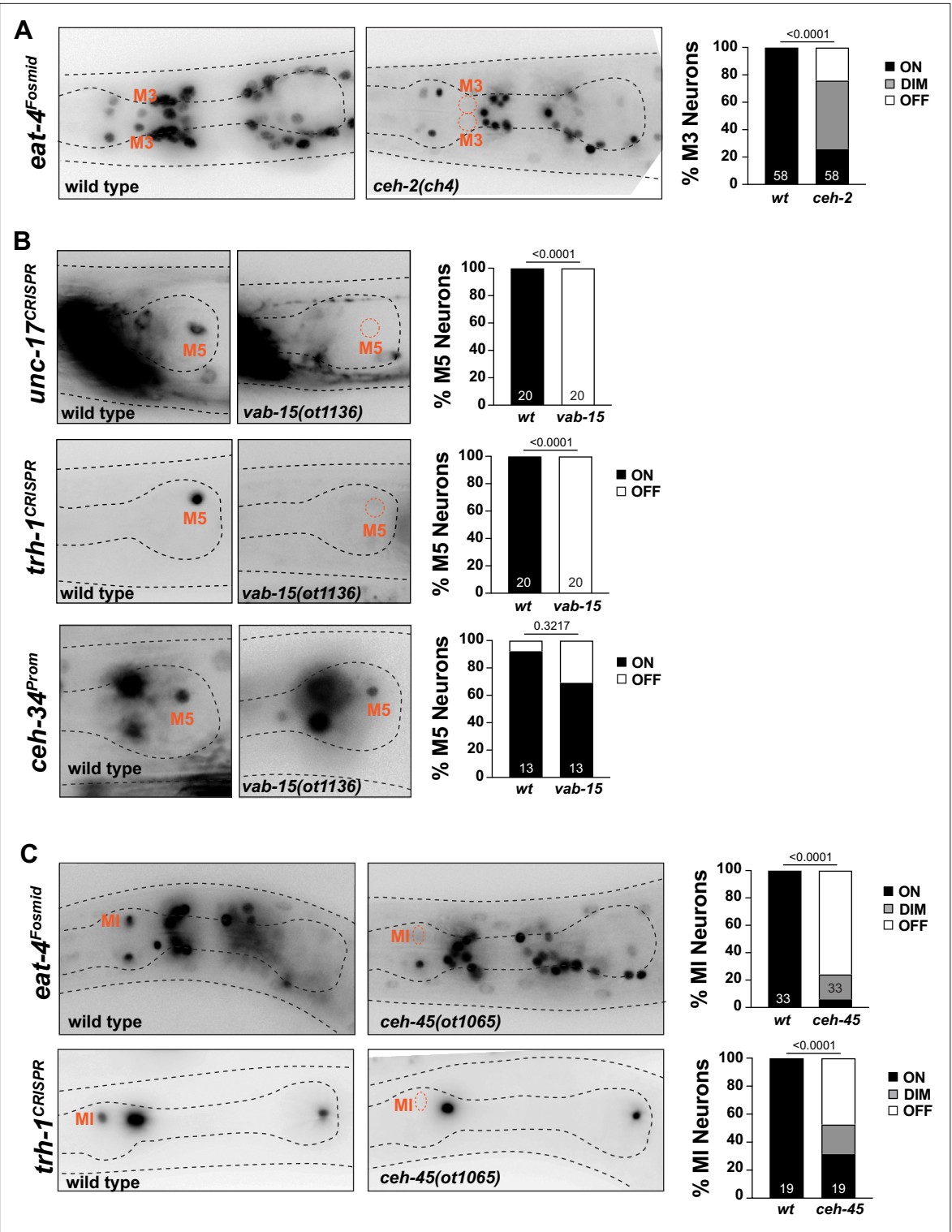

**Figure 11.** Other homeobox genes involved in specifying distinct pharyngeal neuron types. (**A**) *ceh-2* affects M3 neuron differentiation. Representative images and quantification are shown. Reporter gene is *eat-4* (*otIs388*). Animals scored at the L4 stage. Statistical analysis was performed using chi-square test. N is indicated within each bar and represents number of neurons scored. (**B**) *vab-15* affects M5 neuron differentiation. Representative images and quantification are shown. Reporter genes used are *ceh-34* (*stIs10447*) and CRISPR/Cas9-engineered alleles for *unc-17* (*ot907*) and *trh-1* (*syb4421*). Animals were scored at the L4 stage. Statistical analysis was performed using Fisher's exact test. N is indicated within each bar and represents number of neurons scored. (**C**) *ceh-45* affects MI neuron differentiation. Representative images and quantification are shown. Reporter genes used are

*Figure 11 continued on next page*

*Figure 11 continued*

*eat-4* (*otIs388*) and CRISPR/Cas9-engineered allele for *trh-1* (*syb4421*). Animals were scored at the L4 stage. Statistical analysis was performed using Fisher's exact test. N is indicated within each bar and represents number of neurons scored.

The online version of this article includes the following figure supplement(s) for figure 11:

**Figure supplement 1.** Homeobox mutant alleles.

**Figure supplement 2.** Homeobox misexpression experiments.

pharyngeal neurons (*Aspöck et al., 2003*; *Feng and Hope, 2013*; *Mörck et al., 2004*; *Ramakrishnan and Okkema, 2014*; *Rauthan et al., 2007*; *Ray et al., 2008*; *Zhang et al., 2014*), yet no common theme emerged from these studies. We have shown here that a Sine oculis ortholog, *ceh-34,* orchestrates the terminal differentiation program of all pharyngeal neurons. CEH-34 appears to act as a terminal selector of pharyngeal neuron identity, as inferred from its requirement to initiate terminal neuronal differentiation programs in all pharyngeal neurons (without affecting pan-neuronal identity, a unifying trait of all terminal selectors), as well as its continuous role in maintaining the differentiated state (another defining trait of terminal selectors). The aspects of the differentiation program that we consider here, and found to be under control of *ceh-34,* include anatomical (axon outgrowth and synapse formation), molecular, and functional features. Molecular and functional features affected by *ceh-34* range from neuron-neuron communication to presumptive sensory functions to the intriguing function of enteric neurons as potential regulators of microbial colonization. There is good reason to believe that CEH-34 controls these diverse phenotypic identity features in a direct manner, that is, it may not act through intermediary factors. CEH-34 is among the many *C. elegans* transcription factors whose binding sites have been determined in vitro through protein binding microarrays (*Narasimhan et al., 2015*) and a phylogenetic footprinting pipeline reveals that these motifs are significantly enriched in the single cell transcriptome of most pharyngeal neuron classes (*Glenwinkel et al., 2021*). This is in accordance with CEH-34 being a shared terminal selector component of all pharyngeal neuron classes.

Within the nervous system, the selectivity of *ceh-34* expression in all pharyngeal neurons is remarkable – based on extensive gene expression pattern analysis, including recent scRNA data, there is no other transcription factor that so selectively and comprehensively defines all pharyngeal, but no non-pharyngeal neuronal cell types. We found that the key determinant of this expression is the organ selector PHA-4 (*Gaudet and Mango, 2002*; *Horner et al., 1998*; *Kalb et al., 1998*; *Mango et al., 1994*), which is expressed earlier in development to act both as a pioneer factor (*Hsu et al., 2015*) and to induce the expression of a number of different terminal selectors for different tissue types within the foregut – the *ceh-34* gene for all neurons (this paper), the bHLH transcription factor *hlh-6* for pharyngeal gland cells (*Smit et al., 2008*) and the Nk-type homeobox gene *ceh-22* for pharyngeal muscle identity (*Vilimas et al., 2004*). Given the continuous expression of PHA-4 throughout

| homeobox gene | | enteric neuron class | | | | | | | |
|---|---|---|---|---|---|---|---|---|---|
| *C. elegans* | Vertebrate homolog | I1 | I2 | I3 | M3 | M4 | M5 | MI | NSM |
| *ceh-34* | SIX1/2 | ▓ | ▓ | ▓ | ▓ | ▓ | ▓ | ▓ | ▓ |
| *unc-86* | BRN3 | ▓ | | | | | | | ▓ |
| *ttx-3* | LHX2/9 | | | | | | | | ▓ |
| *ceh-2* | EMX1/2 | | | ▓ | | | | | |
| *ceh-14* | LHX3/4 | | ▓ | | | | | | |
| *ceh-45* | GSC | | | | | | | ▓ | |
| *ceh-28* | NKX2 | | | | | ▓ | | | |
| *zag-1* | ZEB1/2 | | | | ▓ | | | | |
| *vab-15* | MSX1/2 | | | | | | ▓ | | |
| *pros-1* | PROX1/2 | | | ▓ | | | | | |

**Figure 12.** Summary of homeobox gene codes involved in pharyngeal neuron identity specification. Shown here are homeobox genes for which an involvement in pharyngeal neuron differentiation has been shown, as well as the 8 (of a total of 14) pharyngeal neuron classes for which a homeobox regulator *besides ceh-34* has been identified to date. Each neuron expresses additional homeobox genes (resulting in neuron type-specific combination of homeobox genes; *Reilly et al., 2020*), but the function of these additional genes remains to be characterized.

postembryonic life, it is conceivable that PHA-4 acts in a regulatory feedforward motif configuration, where it not only induces tissue-type terminal selectors (*ceh-34, hlh-6, ceh-22*), but then also collaborates with them to induce and maintain terminal differentiation batteries.

Our work indicates that CEH-34 is a shared component of neuron type-specific terminal selector complexes, such that CEH-34 interacts with a distinct set of at least eight homeodomain cofactors to impose unique features in distinct pharyngeal neuron types. As is the case for CEH-34 target sites, the in vitro-determined binding sites for several of these homeodomain cofactors display a phylogenetically conserved enrichment in the respective neuron type-specific gene batteries (*Glenwinkel et al., 2021*). For example, CEH-34 and UNC-86 binding sites are co-enriched in the I1 neuronal transcriptome, CEH-34 and CEH-14 binding sites in the I2 neuronal transcriptome, CEH-34 and CEH-2 in the I3 transcriptome, CEH-34 and CEH-45 in the MI transcriptome, and VAB-15 and CEH-34 in the M5 transcriptome (*Glenwinkel et al., 2021*). The interactions of CEH-34 with its various collaborating factors is likely highly dependent on the *cis*-regulatory architecture of individual target genes. We infer this from *ceh-34* mutant phenotypes, which, depending on target gene, can be fully or partially penetrant (i.e. not all animals affected), or fully or partially expressive (i.e. 'dimming' of target gene expression), or a combination of both. Depending on the number, affinity, and arrangement of binding sites for individual factors, CEH-34 and its individual cofactors may each have a more or less pronounced role in the regulation of individual target genes.

We found that the ectopic expression of pharyngeal homeobox genes reveal a limited capacity to respecify identity features of pharyngeal neuron (*Figure 11—figure supplement 2*). This is a likely reflection of our incomplete knowledge of the entire set of collaborating factors and possibly also a reflection of the difficulties associated with overriding endogenous terminal differentiation programs by ectopic expression of drivers of alternative fates (*Patel and Hobert, 2017*).

Taken together, our findings not only reveal a common terminal selector-based regulatory logic for how a self-contained, enteric nervous system acquires its terminal differentiated state. They also corroborate two themes that have emerged from recent studies, primarily in *C. elegans* (and also emerging in other systems):

1. Homeobox genes – and more specifically, combinatorial codes of homeobox genes – are prominently employed in neuron identity specification throughout all neurons of the nervous system, as exemplified here in the context of the enteric nervous system, the so-called 'second brain' of animals (*Gershon, 1998*).
2. A number of identity-specifying terminal selectors, such as CEH-34, are expressed in synaptically connected neurons, suggesting they may specify the assembly of neurons into functional circuitry. It is presently unclear how prominent such a connectivity theme is. We observed a few cases of such putative 'circuit organizer' transcription factors in the non-pharyngeal nervous system (*Berghoff et al., 2021*; *Pereira et al., 2015*) and there are some striking potential examples in vertebrates (*Brunet and Pattyn, 2002*; *Dauger et al., 2003*; *Ha and Dougherty, 2018*; *Ruiz-Reig et al., 2019*; *Sokolowski et al., 2015*). This present study provides an extreme example of this. An entire set of synaptically connected neurons (the worm's enteric nervous system) is specified by a single transcription factor (CEH-34), which apparently helps these neurons to become assembled into functional circuitry. In each pharyngeal neuron type, CEH-34 pairs up with different homeodomain proteins to diversify pharyngeal neurons into distinct identities. Following Dobzhansky's dictum that 'nothing in biology makes sense except in the light of evolution' (*Dobzhansky, 1964*), we speculate that the pharyngeal nervous system may have derived from a homogenous set of interconnected, identical neurons, all specified by *ceh-34*, which may have regulated a homophilic adhesion molecule that functionally linked these ancestral neurons. The more complex, present-day circuitry may have evolved through the eventual partnering of CEH-34 protein with distinct sets of homeodomain proteins that diversified neuronal identities, connectivity, and function in the pharyngeal nervous system.

Arguing for a conserved function of Six homeodomain factors in enteric nervous system differentiation is the observation that in flies, the Sine oculis paralog Optix is indeed expressed in the frontal ganglion (*Seo et al., 1999*), which constitutes the nervous system of insect foreguts (*Hartenstein, 1997*). In the context of studying Sine oculis function in the *Drosophila* corpus cardiacum, it was also noted that the entire stomatogastric ganglion, that is, the entire enteric nervous system (of which the frontal ganglion is a part) does not form in Sine oculis mutants (*De Velasco et al., 2004*). In sea urchin, pharyngeal neurons also express, and require for their proper development, the Sine oculis paralog

Six3 (*Wei et al., 2011*). Other than an early report of Six2 expression in the mouse foregut region (*Ohto et al., 1998*), the expression and function of Sine oculis orthologs in vertebrate enteric nervous systems has, to our knowledge, not yet been examined.

Notably, another homeobox gene appears to have a critical and very broad function in vertebrate enteric nervous system development that is akin to the broadness of *ceh-34* function in *C. elegans*. The paired-type homeobox gene Phox2b is expressed in enteric nervous system precursors and required early in development for the generation of all enteric ganglia (*Pattyn et al., 1999*; *Tiveron et al., 1996*). While Phox2b is also continuously expressed throughout the adult enteric nervous system (*Corpening et al., 2008*; *Drokhlyansky et al., 2020*; *Morarach et al., 2021*), its function in terminal differentiation and perhaps even maintenance of enteric neurons identity remains to be examined, for example, via temporally controlled, postdevelopmental knock-out in juvenile or adult stage animals. If the analogy to *ceh-34* holds, the enteric neurons of such animals may lose their differentiated state. Remarkably, additional homeobox genes have recently been noted to show highly selective expression patterns within the vertebrate enteric nervous system, effectively discriminating distinct neuronal subtypes (*Memic et al., 2018*). One of them, the Meis ortholog Pbx3, has been confirmed to play an important role in the postmitotic specification of distinct enteric neuron types (*Morarach et al., 2021*). Hence, it appears that the overall logic of a pan-enteric homeobox gene, cooperating with cell type-specific homeobox genes, may be conserved from worms to vertebrates.

Another evolutionary perspective of our findings considers the origins of the enteric nervous system, and maybe nervous systems as a whole. Based on a number of anatomical and functional features, it has been proposed that enteric nervous systems preceded, and then paralleled the emergence of centralized nervous systems of bilaterian animals (*Furness and Stebbing, 2018*; *Gilbert, 2019*; *Klimovich and Bosch, 2018*). This argument is bolstered by considering a number of features of the enteric, that is, pharyngeal nervous system of *C. elegans*: (1) its polymodality (sensory + inter + motor neuron) of most pharyngeal neurons, (2) its innervation of what is essentially a single sheath of myoepithelial cells, a proposed feature of primitive nervous systems (*Mackie, 1970*), (3) its simple immune functions (also thought to be a feature of primitive neurons, e.g. in hydra; *Klimovich et al., 2020*; *Klimovich and Bosch, 2018*), and (4) the relatively indiscriminate synaptic cross-innervation patterns among pharyngeal neurons (*Cook et al., 2020*). If pharyngeal neurons indeed resemble a more primitive, ancestral state of neurons, our observation that CEH-34 acts as a terminal selector in these neurons would point to the ancient nature of (1) a terminal selector-type logic of neuronal identity specification and (2) the deployment of a homeobox gene in such function. Sine oculis homologs appear to be employed broadly in sensory neuron specification across animal phylogeny, even in the most basal metazoan (*Jacobs et al., 2007*). CEH-34/Sine oculis may represent an ancestral determinant of neuronal cell types.

## Materials and methods
### *C. elegans* strains
Worms were grown at 20°C on nematode growth media (NGM) plates seeded with *E. coli* (OP50) bacteria as a food source. The wild-type strain used is Bristol N2. A complete list of strains used in this study can be found in *Supplementary file 1*.

### Generation of deletion alleles
Mutant alleles for the *ceh-34, ceh-45, vab-15, ceh-7, ceh-53, ceh-79, and eya-1* genes (schematized in *Figure 2* and *Figure 11—figure supplement 1*) were generated by CRISPR/Cas9 genome engineering as described (*Dokshin et al., 2018*). A deletion of the full locus was generated using two crRNAs and an ssODN donor. Sequences are as follows:

> *ceh-34(ot1014)*: crRNAs (cgacaagaggacgacgctct and ttattctaatggtcttgagg), ssODN (gcgacatt cactgggggacgacaagaggacgacgccaagaccattagaataactttaactatattttg).
> *ceh-34(ot1188)* and *ceh-34(ot1189)* were generated the same way as *ceh-34(ot1014)* and are molecularly identical. The difference is that *ot1188* was generated in the background of *flr-2(syb4861)* and *ot1189* was generated in the background of *htrl-1(syb4895)* because these loci are very closely linked to *ceh-34*.

*ceh-45(ot1065)*: crRNAs (taggccaccgatacaagcag and tccgccagagaccggtcggg), ssODN (aact gaaattcgaaattctaggccaccgatacaaggaccggtctctggcggattactgtagccgtttggg).
*vab-15(ot1136)*: crRNAs (ggtcaacacatctgcttata and ttgtgaaaagcgtaatactt), ssODN (agcgcgtg gtgttatattggtcaacacatctgctttattacgcttttcacaatatttttatggactaacca).
*ceh-7(ot1138)*: crRNAs (ccccttgtactgacaattga and tgatcaggaatttgctctcg), ssODN (cgaaacga aacgggcggcccccttgtactgacaatgagcaaattcctgatcatctgacactttttccagac).
*ceh-53(ot1066)*: crRNAs (gcggcgcttccgggactctg and gaaatcaggggcaaacttgg), ssODN (gctccatc agaaaaaggggcggcgcttccgggactagtttgcccctgatttcgaatatttatgtgaaaaa).
*ceh-79(ot1067)*: crRNAs (aagaagaaccgacgaaccca and cacccccgaactgtgttcac), ssODN (aactcctg tctctccttcgatgatcttttccatggcactggacacatatctttaacttttccgatgtgta).
*eya-1(ot1197)*: crRNAs (ttttgtacgagtgactcagt and acacctgtatctctgcggggg), ssODN (cggtcgtc agattggtagccctccaaaatcccactcgcagagatacaggtgttcaaaatcggggtgaaga).

With the exception of *ceh-34(ot1014)*, *ceh-34(ot1188)* and *ceh-34(ot1189)* animals, all other null mutant alleles are homozygous viable. *ceh-45(ot1065)* and *vab-15(ot1136)* are slow growing and at least *ceh-45(ot1065)* animals also display a partially penetrant embryonic lethality.

## Generation of reporter knock-ins

The *ceh-34* locus was tagged with *mNG::3xFLAG::AID* to generate *ceh-34(ot903)*. The AID sequence was amplified and inserted into the pDD268 vector (*mNG::SEC::3xFLAG*) (**Dickinson et al., 2015**) to generate the plasmid pUA77 (*ccdB::mNG::SEC::3xFLAG::AID ccdB*; **Aghayeva et al., 2021**). The construct contains a self-excising drug selection cassette (SEC) and was used for SEC-mediated CRISPR insertion of *mNG::3xFLAG::AID* right before the stop codon of *ceh-34* as described in **Dickinson et al., 2015**. The guide RNA used targets the following sequence: ttattctaatggtcttgagg.

The *pha-4* locus was tagged with *gfp* at its 3'end to generate *pha-4(ot946)*, using Cas9 protein, tracrRNA, and crRNA from IDT, as previously described (**Dokshin et al., 2018**). One crRNA (attg gagatttataggttgg) and an asymmetric double-stranded *gfp-loxP-3xFLAG* cassette, amplified from a plasmid, were used to insert the fluorescent tag at the C-terminal.

Reporter alleles for *flp-5(syb4513)*, *flp-28(syb3207)*, *flr-2(syb4861)*, *htrl-1(syb4895)*, *ser-7(syb4502)*, *rig-3(syb4763)*, *rig-6(syb4729)*, *trh-1(syb4421)*, and *trhr-1(syb4453)* were generated by CRISPR/Cas9 to insert an *SL2::GFP::H2B* cassette at the C-terminus of the respective gene. For the *unc-17(syb4491)* allele *T2A::GFP::H2B* was inserted at the C-terminus. For the *kin-36(syb4677)* locus, a *GFP::HIS::SL2* sequence was inserted at the N-terminus. These strains were generated by Sunybiotech and are listed in **Supplementary file 1**.

## Generation of transgenic reporter strains

To generate *otIs762(ceh-34prom::TagRFP)* a 3720 bp PCR fragment containing the whole *ceh-34* intergenic region plus the first 2 exons and 2 introns was amplified from N2 genomic DNA and cloned into a TagRFP vector using Gibson Assembly (NEBuilder HiFi DNA Assembly Master Mix, Catalog # E2621L). The following primers were used: aatgaaataagcttgcatgcctgcaTGTTTATTTTCTATGTAATTTCT AATAAAGTCCC and cccgggggatcctctagagtcgacctgcaCTGAAAGTTGAAATATAGAATTTTTAATTTTT TTTTTTTG. The resulting construct was injected as a simple extrachromosomal array (50 ng/µl) into *pha-1(e2123)* animals, using a *pha-1* rescuing plasmid (pBX, 50 ng/µl) as co-injection marker. A representative line was integrated into the genome with gamma irradiation and backcrossed four times.

To generate *otIs785(ceh-34prom::GFP::CLA-1)*, a 3720 bp PCR fragment containing the whole *ceh-34* intergenic region plus the first 2 exons and 2 introns was amplified from N2 genomic DNA and cloned into PK065 (kindly shared by Peri Kurshan) using Gibson Assembly (NEBuilder HiFi DNA Assembly Master Mix, Catalog # E2621L). The following primers were used: gattacgccaagcttgcatg cTGTTTATTTTCTATGTAATTTCTAATAAAGTC and gttcttctcctttactcatcccgggCTGAAAGTTGAAATA TAGAATTTTTAATTTTTTTTTTTTTG. The resulting construct was injected at 7 ng/µl together with *ceh-34prom::TagRFP* (50 ng/µl) (see above) and *rol-6(su1006)* as a co-injection marker. A representative line was integrated into the genome with gamma irradiation and backcrossed four times.

## Choice of pharyngeal fate markers

Most fate markers used in this paper were previously described. For several of those, we used previous expression patterns (based on transgenic reporter fusions) as an impetus to then generate reporter

alleles by CRISPR/Cas9 genome engineering (e.g. *flp-5*, *ser-7*; all listed in previous sections). One case warrants specific emphasis: scRNA has shown that the T11F9.12 gene is expressed exclusively in most, if not all pharyngeal neurons, a notion we confirmed with a CRISPR/Cas9 genome-engineered reporter allele. T11F9.12 encodes for a relatively large (736aa), secreted and nematode-specific protein that contains a Pfam-annotated domain (Htrl domain; PF09612) that is, outside nematodes, only found in a bacterial protein HtrL. This bacterial protein currently has no assigned function but is found in a region of LPS core biosynthesis genes which are involved in bacterial immune defense (*Bertani and Ruiz, 2018*). Interestingly, hidden Markov model-based searches in the Panther database reveal a sequence pattern (PTHR21579) along the entire T11F9.12 protein that is otherwise only found in *C. elegans* saposin proteins, which are bona fide immune effector proteins (*Bányai and Patthy, 1998*; *Hoeckendorf et al., 2012*; *Oishi et al., 2009*). We named this protein HTRL-1.

## Temporally controlled CEH-34 protein degradation

We used conditional protein depletion with a modified auxin-inducible degradation system (*C.e.* AIDv2; *Hills-Muckey et al., 2022*). AID-tagged proteins are conditionally degraded when exposed to 5-Ph-IAA in the presence of $_{At}TIR1^{F79G}$. To generate the experimental strain, the conditional allele *ceh-34(ot903[ceh-34::mNG::AID])* was crossed with *cshIs140[rps-28p::TIR1(F79G)]*, which expresses $_{At}TIR1^{F79G}$ ubiquitously. The synthetic auxin analog 5-Ph-IAA was purchased from BioAcademia (#30-003-10) and dissolved in ethanol (EtOH) to prepare 100 mM stock solutions. NGM agar plates with fully grown OP50 bacterial lawn were coated with the 5-Ph-IAA stock solution to a final concentration of 200 µM and allowed to dry overnight at room temperature. To induce protein degradation, synchronized L1 or young adult worms were transferred onto 5-Ph-IAA -coated plates and kept at 20°C. As a control, worms were transferred onto EtOH-coated plates instead. 5-Ph-IAA solutions and experimental plates were shielded from light.

## Microscopy and image analysis

Worms were anesthetized using 100 mM sodium azide ($NaN_3$) and mounted on 5% agarose pads on glass slides. Z-stack images (each ~0.7 µm thick) were acquired using a Zeiss confocal microscope (LSM880) or Zeiss compound microscope (Imager Z2) with the ZEN software. Maximum intensity projections of 2–30 slices were generated with the ImageJ software (*Schindelin et al., 2012*).

Reporter gene expression in different neurons was visualized in wild-type and mutant animals and usually assigned to one of the following categories: 'on' (fluorescence levels comparable to wild-type animals), 'dim' (fluorescence still detectable but much dimmer than wild-type animals), or 'off' (fluorescence not detectable). In cases were fluorescence levels were variable between animals and difference with wild type was not obvious mean fluorescence intensity in each neuron was measured with the ImageJ software.

## Acknowledgements

We thank Chi Chen for generating transgenic lines, Seth Taylor and Michael Gershon for discussion, Kelly Liu, Robert Horvitz, Peri Kurshan, and Matthias Leippe for providing reagents, Steven Cook for providing illustrations and Michael Gershon and members of the Hobert lab for comments on the manuscript. This work was funded by NIH R21NS106843, NIHR01NS039996, and the Howard Hughes Medical Institute.

## Additional information

### Funding

| Funder | Grant reference number | Author |
| --- | --- | --- |
| Howard Hughes Medical Institute | | Oliver Hobert |
| National Institutes of Health | R01 NS039996 | Oliver Hobert |

| Funder | Grant reference number | Author |
| --- | --- | --- |
| National Institutes of Health | R21NS106843 | Oliver Hobert |

The funders had no role in study design, data collection and interpretation, or the decision to submit the work for publication.

## Author contributions

Berta Vidal, Conceptualization, Data curation, Formal analysis, Investigation, Methodology, Visualization, Writing - review and editing; Burcu Gulez, Formal analysis, Investigation; Wen Xi Cao, Conceptualization, Data curation, Formal analysis, Investigation, Visualization, Writing - review and editing; Eduardo Leyva-Díaz, Molly B Reilly, Tessa Tekieli, Formal analysis, Investigation, Visualization; Oliver Hobert, Conceptualization, Funding acquisition, Project administration, Supervision, Writing - original draft

## Author ORCIDs

Eduardo Leyva-Díaz  http://orcid.org/0000-0001-6750-9168
Molly B Reilly  http://orcid.org/0000-0002-7180-7763
Oliver Hobert  http://orcid.org/0000-0002-7634-2854

## Decision letter and Author response

Decision letter https://doi.org/10.7554/eLife.76003.sa1
Author response https://doi.org/10.7554/eLife.76003.sa2

# Additional files

## Supplementary files

• Supplementary file 1. Strain list. This file provides a list of all *Caenorhabditis elegans* strains used in this study.

• Transparent reporting form

• Source data 1. This file provides all the primary animal scoring data.

## Data availability

All data generated or analysed during this study are included in the manuscript and supporting files.

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
