## [Editor Report]

This paper marks a significant advance in understanding the transcriptional control of neural cell fate and connectivity. The authors show that a single homeodomain transcription factor has a central role in specifying the diverse neuron types of the enteric nervous system of the *C. elegans* pharynx. By linking cell fates across a single, largely self-contained circuit, these studies support the emerging idea that transcriptional control can link cell fate to circuit connectivity and function.

---

## [Decision Letter]

**Decision letter after peer review:**

Thank you for submitting your article "The enteric nervous system of *C. elegans* is specified by the Sine Oculis-like homeobox gene *ceh-34*" for consideration by *eLife*. Your article has been reviewed by 3 peer reviewers, including Douglas Portman as Reviewing Editor and Reviewer #1, and the evaluation has been overseen by Piali Sengupta as the Senior Editor. The following individuals involved in review of your submission have agreed to reveal their identity: Carlos Díaz-Balzac (Reviewer #1 co-reviewer); Stefan Thor (Reviewer #3).

As you will see, all three reviewers find your work to be of significant interest. However, there are a number of points that need to be addressed for the claims of your paper to be fully supported. Further details and comments can be found in the individual reviews below; we hope these will be helpful.

Essential revisions:

1) Reviewers 1 and 3 note some concerns about the execution and interpretation of the post-embryonic depletion experiments. Please address these by further characterizing the phenotypes of ceh-34 depletion in larvae (particularly architecture and/or connectivity) and by determining the consequences of ceh-34 depletion in late larvae and/or adults.

2) Reviewers 1 and 2 have significant concerns about your use of the term "enteric". Previous work (including WormAtlas, the canonical source) uses "enteric" to describe muscles and neurons involved in defecation behavior, and the pharynx is considered to be different from the gut. Please provide a more compelling rationale for redefining "enteric", or instead use "pharyngeal" in the title and main text and speculate about "enteric" in the Discussion.

3) Reviewers 1 and 3 point out the lack of ectopic/mis-expression experiments. At a minimum, please test the sufficiency of ceh-34 expression for promoting pharyngeal neuron characteristics; ideally, this would be done by testing combinatorial sufficiency using both ceh-34 and one or more of its partners.

4) Reviewer 3 makes additional points about the rigor of the combinatorial code studies (point 2) that can be addressed by examining additional markers and/or modifying the text.

5) All three reviewers have comments about the significance of the "dimming" of reporter expression that is seen in a number of experiments. These can be addressed with modification of the text and figures.

6) Reviewer 2 notes issues regarding ceh-34 genetics (points 3 and 6) and methods (point 7) that should be addressed in a resubmission.

7) As noted by Reviewer 2, the paper would be strengthened by reporting the functional consequences of ceh-34 loss. As this could be seen as being beyond the scope of the paper, we do not consider these revisions essential, but please consider addressing this point by, for example, disrupting ceh-34 function specifically in M4 (Reviewer #2, point 5).

*Reviewer #1 (Recommendations for the authors):*

1. The authors showed that postembryonic depletion of ceh-34 results in lack of expression of eat-4/VgluT and unc-17/VAChT, but do not comment on whether it also affects pharyngeal nervous system architecture. Asking whether ceh-34 is required postembyronically for the maintenance of architecture/morphology/connectivity would be useful.

2. Figures4C and 5C overstate the requirement for ceh-34 in gene expression. For a number of reporters (glr-2 and several neuropeptides) expression is not completely abolished in ceh-34 mutants, as the figure suggests. Lightly shading the appropriate neurons on the right side would be a more accurate representation of the results. (Also, I wonder whether the residual expression of some of these reporters might depend on ceh-33 – it might be useful for the authors to comment/speculate here.)

3. Authors state that "Constitutive auxin exposure does not phenocopy the larval arrest phenotype of ceh-34 null mutants" (p. 13). I assume "constitutive" means beginning at hatching (Figure 7)? I wouldn't really consider this constitutive, as embryonic ceh-34 function would be unaffected. It would be interesting to know whether exposing mothers (perhaps starting in L3/L4) to auxin would lead to more severe effects in their progeny. Along these lines, it would also be interesting to know how long the requirement for ceh-34 persists. Does auxin treatment of adults have an effect on pharyngeal neuron gene expression and/or pharynx function?

4. None of the studies in this paper explore potential instructive roles of ceh-34 using ectopic expression experiments. Would expression of ceh-34 (perhaps together with another homeodomain partner and/or eya-1) in another neuron type activate pharyngeal targets? Would expression in nearby neurons bring about synaptic connectivity? I realize these questions might not be as straightforward to ask as they seem, but some attempt to address them would be useful. At a minimum it would be informative for the authors to speculate about this in the Discussion.

*Reviewer #2 (Recommendations for the authors):*

I encourage authors to carefully consider terminology used and to condense the data presentation. The manuscript would also be strengthened by some experiments that speak to the functional consequences of ceh-34 mutation.

Major concerns:

(1) The authors should carefully consider calling the pharyngeal nervous system the 'enteric' nervous system. This seems to be a discussion point, but the authors have elevated it to being a fact.

(2) The authors look at many aspects of cell fate in the pharyngeal nervous system but they do not consider one that has been previously reported: programmed cell deaths (PCDs) in the lineages that generate the pharyngeal nervous system. Does deletion of ceh-34 cause widespread defects in PCD and the appearance of extra neurons?

(3) An allele of ceh-34 was previously isolated by a screen to identify factors that specified the cell death of the M4 neuron (Hirose et al., 2010, PMID 17942697). This mutant allele is reported to carry a mutation that disrupts a splice-acceptor sequence. The authors refer to this allele as a missense allele – is this correct? The authors refer to this allele as a hypomorphic allele because it does not display the L1 arrest defect that is caused by deletion of ceh-34. It is important to show that this allele does indeed have residual function; the interpretation of data shown in figures 10 and 11 hinges upon this issue.

(4) Quantification of gene expression defects in Figures 11 and 12 has a qualitative component. How are cells assigned to the 'dim' category? The method should be more clearly explained or gene expression should be quantified.

(5) Only one pharyngeal neuron – M4 – is required for viability. The authors' model strongly suggests that manipulating ceh-34 in this neuron will either cause or rescue the observed L1 arrest defect. If possible, this should be tested. In a similar vein, there are conditions that allow the growth of M4-ablated animals, and the authors' model predicts that these conditions will bypass a requirement for CEH-34. This could also be tested.

(6) The authors should determine whether the M4-specific expression of CEH-28 requires CEH-34. If CEH-28 expression persists in the absence of CEH-34, the authors should revise their discussion to more clearly address the possibility that CEH-34-independent mechanisms determine neuronal cell fates in the pharyngeal nervous system.

(7) In eya-1 mutants, cell bodies are mispositioned. How do the authors know the identities of the cells they are counting if they cannot rely on nuclear position or cell morphology?

*Reviewer #3 (Recommendations for the authors):*

1) Figure 7: They claim that ceh-34 is required to maintain differentiated features. But the auxin treatment, and the degradation of ceh-34, is conducted immediately after the worms have hatched into L1. Can they treat the worms as adults instead, and/or L2-L4, and score e.g., eat-4 expression? Moreover, they claim that ceh-34 is required to maintain differentiated features. But they only test two markers, eat-4 and unc-17, and eat-4 is only affected in some cells and unc-17 is only reduced. What happens to all the other 20-30 markers that were analysed in the constitutive mutant allele? This part of the manuscript is quite interesting but does not hold the same rigour as the rest of the study.

2) The mutant analysis of the other TFs is not comprehensive, with few markers analysed and not all TFs analysed. For unc-86 it has only a minor effect (dimmed expression of cat-1). They refer to a previous study where unc-86 was found to "affect many but not all NSN marker genes"; which? For ttx-3, they refer to the same previous study, where it was found to be a "regulator of NSM differentiation", but the phenotype and markers are not outlined. ceh-14 has no effect on its own and ceh-2 shows only dimming of markers in I3. Basically, the combinatorial coding halters in its description and in the observed phenotypes.

3) On the same topic, there is no discussion of what complete loss of a marker versus "dimming" means. Specifically, if two TFs are acting in a combinatorial code to dictate a specific neuronal cell-fate, possibly acting cooperatively on the same set of downstream genes (enhancers), but one shows complete loss of marker expression and the other "dimming", what does this mean with regards to TFs-enhancer logic?

4) They only test the necessity of the genes. While this is certainly the most important issue, many genes may be necessary for a cell's differentiation without really governing its fate. For instance, eye-specific mutant screens in *Drosophila* identified some 9% of all genes as being necessary for eye development (~1,500 genes). However, only a few of these can act to dictate eye development. The study would be greatly strengthened by single and combinatorial misexpression experiments, to probe if these TF codes are also sufficient to dictate cell fate in other cells. Combinatorial misexpression has been extensively done in *Drosophila*, zebrafish, chick, mouse and iPSC, with stellar results, and the worm is lagging behind in this space.

[Editors’ note: further revisions were suggested prior to acceptance, as described below.]

Thank you for resubmitting your work entitled "The enteric nervous system of *C. elegans* is specified by the Sine Oculis-like homeobox gene *ceh-34*" for further consideration by *eLife*. Your revised article has been evaluated by Piali Sengupta (Senior Editor) and a Reviewing Editor.

The original reviewers have seen and discussed your responses to the initial reviews. All reviewers agree that you have addressed the majority of their concerns and that the paper is nearly ready for publication. However, there are two remaining issues we would like you to address.

1) The reviewers appreciate the detailed rationale you have provided regarding the use of the term "enteric." They largely agree with your point that it is "justified to call the pharyngeal nervous system an enteric nervous system." However, in your paper, and particularly in the title, you refer to this as "*the* enteric nervous system" of *C. elegans*. This seems to imply that DVB and AVL, which innervate enteric muscles, should not be considered enteric. Is there a way for you to make the point that the pharyngeal nervous system should be considered part of the enteric nervous system without implying that it is its sole component?

2) Regarding the new misexpression experiments, the reviewers find the results of these studies interesting and feel that they will be useful to others in the field. Some of the reviewers find it surprising that you observed limited effects in these experiments, since in other systems, researchers have obtained robust ectopic generation of various neuronal sub-types by co-misexpression of 2-4 TFs. Further, one reviewer notes that studies in other systems have found that "TF co-misexpression can simply add ectopic neurotransmitter expression on top on the already existing one, hence creating a mixed cell fate," which means that it is not always necessary to "override the endogenous differentiation program," as may be the case in your studies. To address these points, the reviewers would like to you include the new misexpression data in the manuscript, perhaps in an additional supplementary figure, and discuss this issue more thoroughly in the Discussion.

---

## [Author Response]

Essential revisions:1) Reviewers 1 and 3 note some concerns about the execution and interpretation of the post-embryonic depletion experiments. Please address these by further characterizing the phenotypes of ceh-34 depletion in larvae (particularly architecture and/or connectivity) and by determining the consequences of ceh-34 depletion in late larvae and/or adults.

The postembryonic depletion experiments have been very substantially improved: (a) we have now done the depletion from adult, rather than larval stage animals, and maintenance defects are confirmed; (b) we have tested additional cell fate markers (total of four instead of the original two); (c) we have undertaken a functional analysis, showing that adult (or larval) *ceh-34* depletion results in (expected) pharyngeal pumping defects; (d) we have shown that adult (or larval) *ceh-34* depletion results in synaptic disorganization. These are striking results that forcefully illustrate the importance of *ceh-34* not only in circuit development, but also circuit maintenance.

2) Reviewers 1 and 2 have significant concerns about your use of the term "enteric". Previous work (including WormAtlas, the canonical source) uses "enteric" to describe muscles and neurons involved in defecation behavior, and the pharynx is considered to be different from the gut. Please provide a more compelling rationale for redefining "enteric", or instead use "pharyngeal" in the title and main text and speculate about "enteric" in the Discussion.

Calling pharyngeal neurons enteric neurons is not novel; a recent publication from the Flavell lab has begun to do so (PMID 30580965). Irrespective of this precedent, such naming is in our opinion very well justified and the reasoning is as follows:

(a) As per any animal anatomical textbook definition, the enteric nervous system is the nervous system of the gastrointestinal system.

(b) Again, as per any textbook definition, the foregut is part of the gastrointestinal system.

(c) The pharynx is the worm foregut and hence, the nervous system of the pharynx constitutes an enteric nervous system. (The foregut terminology for the worm pharynx has been used extensively before in the literature and for a good reason: Most animals, including mammals and humans, have a pharynx, which is considered part of the foregut. The only “unusual” thing about worms is that its pharynx is the only part of the foregut, while in other animals (incl. us), the foregut contains the pharynx, plus additional subdivisions, like the esophagus).

(d) As importantly, the classification of the pharyngeal nervous system as an enteric system is also underscored by functional criteria: The two most distinguishing features of an enteric nervous system in animals, namely (i) its autonomous function and (ii) its rhythmic control of peristaltic movement are the defining features of the pharyngeal nervous system as well.

(e) Lastly, the AVL and DVB neurons have also been labeled – justifiably so – enteric neurons before. What those neurons do is to innervate the HINDGUT, not more, not less. By the same token, neurons that innervate the FOREGUT deserve to be called enteric neurons as well – as they actually are across animal phylogeny (as explained above).

We conclude that it is therefore justified to call the pharyngeal nervous system an enteric nervous system.

We have carefully considered this matter and have also double-checked this terminology issue with the world expert of animal enteric nervous systems, our colleague Mike Gershon, who authored the book “The second brain” ( = the enteric nervous system).

We re-iterate these points in our individual response to each reviewer (who appear to be more intrigued than opposed to this) and we also revised the introduction on page 2 to make the point above a little more clearly and to avoid the impression that the pharyngeal nervous system has always been called an enteric nervous system.

3) Reviewers 1 and 3 point out the lack of ectopic/mis-expression experiments. At a minimum, please test the sufficiency of ceh-34 expression for promoting pharyngeal neuron characteristics; ideally, this would be done by testing combinatorial sufficiency using both ceh-34 and one or more of its partners.

We have now done several such experiments: We have co-misexpressed two homeobox genes (*unc-86* and *ttx-3*) that we know to be required for NSM differentiation throughout the pharyngeal nervous system (that already expresses *ceh-34* and its cofactor *eya-1*) and found that each one of three transgenic lines displayed 10-40% penetrant NSM marker ectopic expression but only in two of the 14 pharyngeal neuron classes. Moreover, we have misexpressed the same two NSM regulators (*unc-86*, *ttx-3*) together with *ceh-34* and its cofactor *eya-1* outside the pharyngeal nervous system in a diverse set of ~8 neuron classes (using the *unc-47* driver) and found exceptionally low penetrant effect in two lines (1 animal out of ~50 scored, in two lines).

These very limited defects are entirely expected, based on two confounding factors: (1) In contrast to necessity experiments, where removal of single factors can and does result in differentiation defects, sufficiency require the misexpression of the COMPLETE set of TFs involved in identity specification. We apparently do not know the complete set of regulators for all pharyngeal neurons. (2) As importantly, ectopically expressed TFs need to override the endogenous differentiation program of a neuron. As we have explicitly shown in a paper in *eLife* in 2017 (Patel and Hobert), terminal differentiation does not only involve the activation of a specific gene battery, but also involves the active repression of alternative differentiation programs, likely via chromatin-based mechanisms. Hence, misexpression of a (likely insufficient) combination of TFs is unlikely to result in strong effects. We do mention these negative results and their interpretation now in the Discussion, as requested by the reviewer.

4) Reviewer 3 makes additional points about the rigor of the combinatorial code studies (point 2) that can be addressed by examining additional markers and/or modifying the text.

Reviewer 3 does not question the rigor of the combinatorial analysis, but the extent to which this analysis was done. He stated that we only looked at some potential cofactors and at some terminal markers. That is true. We only analyzed a subset of potential homeobox cofactors and only a limited number of markers, but we ask to please take the scope of our analysis into account: At the end of the day, we analyze 10 homeobox genes in the context of eight (of the 14) different pharyngeal neuron classes. Several of these homeobox genes had never been functionally analyzed before and for several of these neuron classes, no identity regulator had been identified. Given the normal standard of cell fate analysis (most often analyzing one or two genes in one neuron), we think that the extent of our analysis is already quite extensive as is.

However, having defended our analysis as is, we have added more data in the revised version of the manuscript:

1) We added the mutant analysis of the worm homolog of the Prospero homeobox gene, *pros^-1^,* whose function had not previously been characterized at all in the nervous system; we found it to be expressed in the pharyngeal I3 neuron and observed differentiation defects, therefore making it another *ceh-34* collaborator. The addition of this data also meant the addition of another author on this paper (Molly B Reilly), who had been studying *pros^-1^* on the side.

2) We added another marker for *ceh-2* mutant analysis

3) We had included in our analysis only phylogenetically conserved homeobox genes. Out of curiosity, we had also undertaken an analysis of three unusual, non-conserved homeobox genes, each of which expressed in subsets of pharyngeal neurons. We did not observe any defects of null mutant alleles that we generated for these genes, either alone or in combination with the *ceh-34* hypomorphic allele. This data is now added into the manuscript as well.

It is clear that the cofactor analysis is just at its beginning. Our sole purpose here was to show that *ceh-34* does interact with distinct homeobox genes in distinct pharyngeal neuron classes. More analysis clearly is required to fully define the complete set of interacting transcription factors. Given the amount of data already present in the manuscript, we feel that more analysis of these cofactors should be left to future studies.

5) All three reviewers have comments about the significance of the "dimming" of reporter expression that is seen in a number of experiments. These can be addressed with modification of the text and figures.

Addressed, as described in response to individual reviewers.

6) Reviewer 2 notes issues regarding ceh-34 genetics (points 3 and 6) and methods (point 7) that should be addressed in a resubmission.

Yes, fixed.

7) As noted by Reviewer 2, the paper would be strengthened by reporting the functional consequences of ceh-34 loss. As this could be seen as being beyond the scope of the paper, we do not consider these revisions essential, but please consider addressing this point by, for example, disrupting ceh-34 function specifically in M4 (Reviewer #2, point 5).

We now provide such functional analysis. In the original version of the manuscript, we had only described the L1 arrest phenotype of the null allele. In this revised version we have circumvented this arrest phenotype by removing *ceh-34* only from adult animals (using the AID system) and found that in those animals pharyngeal pumping is – as expected from disabling the pharyngeal/enteric nervous system – defective.

Reviewer #1 (Recommendations for the authors):1. The authors showed that postembryonic depletion of ceh-34 results in lack of expression of eat-4/VgluT and unc-17/VAChT, but do not comment on whether it also affects pharyngeal nervous system architecture. Asking whether ceh-34 is required postembyronically for the maintenance of architecture/morphology/connectivity would be useful.

We thank the reviewer for this very good suggestion; our use of the AID allele for cell fate analysis much pre-dated our analysis of anatomy and we never revisited the issue. We have now assessed synaptic organization after larval or adult depletion of CEH-34 and found this to result in the formation of aberrant presynaptic clusters, indicating that CEH-34 is also required to maintain synaptic architecture.

In addition, we have also now assessed the functional consequences of larval or adult CEH-34 depletion and found strong pharyngeal pumping defects, as expected from CEH-34 being continuously required to maintain gene expression in pharyngeal neurons. These results are now shown in Figure 7 and Figure 8-Supplement.

2. Figures4C and 5C overstate the requirement for ceh-34 in gene expression. For a number of reporters (glr-2 and several neuropeptides) expression is not completely abolished in ceh-34 mutants, as the figure suggests. Lightly shading the appropriate neurons on the right side would be a more accurate representation of the results. (Also, I wonder whether the residual expression of some of these reporters might depend on ceh-33 – it might be useful for the authors to comment/speculate here.)

Yes, good point. Light shading implemented now in Figures.

In regard to *ceh-33*, the gene is not expressed in pharyngeal neurons. We think it is more parsimonious to ascribe this residual expression to the activity of cofactors of *ceh-34 –* described later in the manuscript. We now make a statement to this effect on p.8.

3. Authors state that "Constitutive auxin exposure does not phenocopy the larval arrest phenotype of ceh-34 null mutants" (p. 13). I assume "constitutive" means beginning at hatching (Figure 7)? I wouldn't really consider this constitutive, as embryonic ceh-34 function would be unaffected. It would be interesting to know whether exposing mothers (perhaps starting in L3/L4) to auxin would lead to more severe effects in their progeny. Along these lines, it would also be interesting to know how long the requirement for ceh-34 persists. Does auxin treatment of adults have an effect on pharyngeal neuron gene expression and/or pharynx function?

Constitutive actually means exposure from parental generation throughout progeny. The original auxin publication, as well as ensuing studies, shows that auxin also works in embryos, i.e. that auxin-exposed adults deliver auxin to developing progeny in utero. Such constitutive exposure indeed reduces ceh-34::gfp::AID both embryonically and postembryonically. However, we still see clear hints of very dim expression – hence, explaining why we only see partial phenotypes. We have since repeated these experiments with a modified AID system (5-Ph-IAA/TIR1^F79G^) that was recently published, but which still did not completely remove all CEH-34 protein.

With the caveat of incomplete depletion in mind, we have pursued *ceh-34* removal at the adult stage, as the reviewer suggests. This worked essentially as well as the constitutive removal, i.e. we observed that multiple markers (we tested two more markers than in the original submission) fail to be properly maintained upon adult removal of *ceh-34*. We have also analyzed functional consequences of adult *ceh-34* removal and found that, as expected, animals display pharyngeal pumping defects.

4. None of the studies in this paper explore potential instructive roles of ceh-34 using ectopic expression experiments. Would expression of ceh-34 (perhaps together with another homeodomain partner and/or eya-1) in another neuron type activate pharyngeal targets? Would expression in nearby neurons bring about synaptic connectivity? I realize these questions might not be as straightforward to ask as they seem, but some attempt to address them would be useful. At a minimum it would be informative for the authors to speculate about this in the Discussion.

We have now done several ectopic expression experiments: We have co-misexpressed two homeobox genes (*unc-86* and *ttx-3*) that we know to be required for NSM differentiation throughout the pharyngeal nervous system (that already expresses *ceh-34* and its cofactor *eya-1*) and found that each one of three transgenic lines displayed 10-40% penetrant ectopic expression of an NSM marker gene but only in two of the 14 pharyngeal neuron classes. Moreover, we have misexpressed the same two NSM regulators (*unc-86*, *ttx-3*) together with *ceh-34* and its cofactor *eya-1* outside the pharyngeal nervous system in a diverse set of ~8 neuron classes (using the *unc-47* driver) and found exceptionally low penetrant effect in two lines (1 animal out of ~50 scored, in two lines).

These very limited defects are entirely expected, based on two confounding factors: (1) In contrast to necessity experiments, where removal of single factors can and does result in differentiation defects, sufficiency require the misexpression of the COMPLETE set of TFs involved in identity specification. We apparently do not know the complete set of regulators for all pharyngeal neurons. (2) As importantly, ectopically expressed TFs need to override the endogenous differentiation program of a neuron. As we have explicitly shown in a paper in *eLife* in 2017 (Patel and Hobert), terminal differentiation does not only involve the activation of a specific gene battery, but also involves the active repression of alternative differentiation programs, likely via chromatin-based mechanisms. Hence, misexpression of a (likely insufficient) combination of TFs is unlikely to result in strong effects. We do mention these negative results and their interpretation now in the Discussion, as requested by the reviewer.

Reviewer #2 (Recommendations for the authors):I encourage authors to carefully consider terminology used and to condense the data presentation. The manuscript would also be strengthened by some experiments that speak to the functional consequences of ceh-34 mutation.(1) The authors should carefully consider calling the pharyngeal nervous system the 'enteric' nervous system. This seems to be a discussion point, but the authors have elevated it to being a fact.

Calling pharyngeal neurons enteric neurons is not novel; a recent publication from the Flavell lab has begun to do so (PMID 30580965). Irrespective of this precedent, such naming is in our opinion very well justified and the reasoning is as follows:

(a) As per any animal anatomical textbook definition, the enteric nervous system is the nervous system of the gastrointestinal system.

(b) Again, as per any textbook definition, the foregut is part of the gastrointestinal system.

(c) The pharynx is the worm foregut and hence, the nervous system of the pharynx constitutes an enteric nervous system. (The foregut terminology for the worm pharynx has been used extensively before in the literature and for a good reason: Most animals, including mammals and humans, have a pharynx, which is considered part of the foregut. The only “unusual” thing about worms is that its pharynx is the only part of the foregut, while in other animals (incl. us), the foregut contains the pharynx, plus additional subdivisions, like the esophagus).

(d) As importantly, the classification of the pharyngeal nervous system as an enteric system is also underscored by functional criteria:

The two most distinguishing features of an enteric nervous system in animals, namely (i) its autonomous function and (ii) its rhythmic control of peristaltic movement are the defining features of the pharyngeal nervous system as well.

(e) Lastly, the AVL and DVB neurons have also been labeled – justifiably so – enteric neurons before. What those neurons do is to innervate the HINDGUT, not more, not less. By the same token, neurons that innervate the FOREGUT deserve to be called enteric neurons as well – as they actually are across animal phylogeny (as explained above).

We conclude that it is therefore justified to call the pharyngeal nervous system an enteric nervous system.

We have carefully considered this matter and have also double-checked this terminology issue with the world expert of animal enteric nervous systems, our colleague Mike Gershon, who authored the book “The second brain” ( = the enteric nervous system).

In the revised version, we clarify this definitional issue in the Introduction (page 2 and 3).

(2) The authors look at many aspects of cell fate in the pharyngeal nervous system but they do not consider one that has been previously reported: programmed cell deaths (PCDs) in the lineages that generate the pharyngeal nervous system. Does deletion of ceh-34 cause widespread defects in PCD and the appearance of extra neurons?

This is a good question. It so happens that we have inadvertently looked at this issue by checking whether *ceh-34* affects *pha-4* expression. *pha-4* is a marker of all pharyngeal cells. By carefully scoring the number of pha-4prom-positive cells, we observe no significant change in overall number of cells (Figure 2 – Suppl 2). In the revised version, we now mention this on p.7/8.

In terms of specifically looking at neurons: We have not seen extra neurons with our cell-specific markers (no surprise because they are usually off), but we could in theory observe extra neurons with a pan-neuronal marker – and we indeed scored multiple pan-neuronal markers in *ceh-34* null mutants (Figure 3A) – however, unlike in the case of *pha-4*, which is nicely restricted to the pharynx, the number of cells expressing pan-neuronal markers is hard to precisely score because pharyngeal tissue is surrounded by non-pharyngeal neurons. Due to the pharyngeal tissue disorganization in *ceh-34* mutants, it’s hard to say whether certain signals come from pharyngeal or non-pharyngeal cells. Superficially, there is no obvious difference.

(3) An allele of ceh-34 was previously isolated by a screen to identify factors that specified the cell death of the M4 neuron (Hirose et al., 2010, PMID 17942697). This mutant allele is reported to carry a mutation that disrupts a splice-acceptor sequence. The authors refer to this allele as a missense allele – is this correct? The authors refer to this allele as a hypomorphic allele because it does not display the L1 arrest defect that is caused by deletion of ceh-34. It is important to show that this allele does indeed have residual function; the interpretation of data shown in figures 10 and 11 hinges upon this issue.

Thanks for catching this misnomer – the hypomorphic allele that we used is indeed not a missense, but a splice site mutation. But it is definitively a hypomorphic allele. We can infer that it has residual function because of the much weaker phenotypes it has on neuronal cell fate markers compared to the null allele. For example, compare effect on *unc-17* or *eat-4* expression in Figure 10/11 with those of the null (Figure 3).

(4) Quantification of gene expression defects in Figures 11 and 12 has a qualitative component. How are cells assigned to the 'dim' category? The method should be more clearly explained or gene expression should be quantified.

The binning into loss of expression or dim expression is now explained in the methods.

(5) Only one pharyngeal neuron – M4 – is required for viability. The authors' model strongly suggests that manipulating ceh-34 in this neuron will either cause or rescue the observed L1 arrest defect. If possible, this should be tested. In a similar vein, there are conditions that allow the growth of M4-ablated animals, and the authors' model predicts that these conditions will bypass a requirement for CEH-34. This could also be tested.

These are good points. However, we caution that *ceh-34* is also expressed in two pharyngeal muscle (pm1/2) and pharyngeal epithelial cells (e1/e2) that are involved in attaching the pharynx to the mouth. *ceh-34* could potentially function in those cells too.

The bigger picture question that we think the reviewer is getting at (based also on the reviewer’s public review at the beginning) is an assessment of the functional, i.e. behavioral consequences of *ceh-34* removal. In the revised version of the manuscript, we have now addressed this question via AID-system-mediated removal of CEH-34 from young adult animals: these animals display severe defects in pharyngeal pumping, as one would expect from disabling the pharyngeal nervous system. This data is now shown in Figure 7.

(6) The authors should determine whether the M4-specific expression of CEH-28 requires CEH-34. If CEH-28 expression persists in the absence of CEH-34, the authors should revise their discussion to more clearly address the possibility that CEH-34-independent mechanisms determine neuronal cell fates in the pharyngeal nervous system.

We tested *ceh-28* expression, as requested, and found that *ceh-34* affects its expression. Now shown in Figure 6 – Supplement 2.

(7) In eya-1 mutants, cell bodies are mispositioned. How do the authors know the identities of the cells they are counting if they cannot rely on nuclear position or cell morphology?

In our hands, cell body positions were not as badly mispositioned to prevent unambiguous assignment of neuronal identities with markers expressed in limited numbers of cells, neither in the previously available ok allele, nor our own new null allele.

Reviewer #3 (Recommendations for the authors):1) Figure 7: They claim that ceh-34 is required to maintain differentiated features. But the auxin treatment, and the degradation of ceh-34, is conducted immediately after the worms have hatched into L1. Can they treat the worms as adults instead, and/or L2-L4, and score e.g., eat-4 expression? Moreover, they claim that ceh-34 is required to maintain differentiated features. But they only test two markers, eat-4 and unc-17, and eat-4 is only affected in some cells and unc-17 is only reduced. What happens to all the other 20-30 markers that were analysed in the constitutive mutant allele? This part of the manuscript is quite interesting but does not hold the same rigour as the rest of the study.

We have done the experiments suggested by the reviewer:

1) We have now removed *ceh-34* in the adult stage and still observed a failure to maintain the differentiated state; this is indeed a much better experiment than our previous L1 experiments and we are grateful that the reviewer pushed us in this direction.

2) We have done the adult removal not just with the two original markers (ACh/unc-17 and Glu/eat4), but have added two additional markers. Again, maintenance defects are observed. We note that these 4 markers provide a broad coverage of pharyngeal neuronal cell types.

3) Rather than testing more markers, we have undertaken a functional analysis, asking whether *ceh-34* removal in the adult results in expected pharyngeal pumping defects. We found this to be indeed the case.

All the new data is now included in Figure 7.

4) Lastly, we now also add data that shows that synaptic organization becomes disorganized upon adult removal of *ceh-34*.

Taken together, these findings clearly demonstrate a continuous requirement for CEH-34 even in adult animals.

2) The mutant analysis of the other TFs is not comprehensive, with few markers analysed and not all TFs analysed. For unc-86 it has only a minor effect (dimmed expression of cat-1). They refer to a previous study where unc-86 was found to "affect many but not all NSN marker genes"; which? For ttx-3, they refer to the same previous study, where it was found to be a "regulator of NSM differentiation", but the phenotype and markers are not outlined. ceh-14 has no effect on its own and ceh-2 shows only dimming of markers in I3. Basically, the combinatorial coding halters in its description and in the observed phenotypes.

Yes, we only analyzed a subset of potential homeobox cofactors and only a limited number of markers, but we ask the reviewer to please take the scope of our analysis into account: At the end of the day, we analyze 10 homeobox genes in the context of eight (of the 14) different pharyngeal neuron classes. Several of these homeobox genes had never been functionally analyzed before and for several of these neuron classes, no identity regulator had been identified. Given the normal standard of cell fate analysis (most often analyzing one or two genes in one neuron), we think that the extent of our analysis is already quite extensive as is. However, having defended our analysis as is, we have added more data in the revised version of the manuscript:

1) We added the mutant analysis of the worm homolog of the Prospero homeobox gene, *pros^-1^,* whose function had not previously been analyzed at all in the nervous system; we found it to be expressed in the pharyngeal I3 neuron and observed differentiation defects, therefore making it another *ceh-34* collaborator. The addition of this data also meant the addition of another author on this paper, who had been studying *pros^-1^* on the side.

2) We added another marker for *ceh-2* mutant analysis

3) We had included in our analysis only phylogenetically conserved homeobox genes. Out of curiosity, we had also undertaken an analysis of three unusual, non-conserved homeobox genes, each of which expressed in subsets of pharyngeal neurons. We did not observe any defects of null mutant alleles that we generated for these genes, either alone or in combination with the *ceh-34* hypomorphic allele. This data is now added into the manuscript as well (text and Figure 11-Suppl.1).

In regard to the specific nature of the defects that we observe and that the reviewer comments on: In a previous analysis of the NSM neurons, we had shown that two genes, *unc-86* and *ttx-3* have synergistic effects on proper NSM neuron differentiation. Several differentiation markers were only partially affected in the single mutants, but a much stronger, if not complete loss was observed in the double mutant. We add *ceh-34* here into the picture by showing that much like *ttx-3*, a *ceh-34* hypomorph also synergizes with *unc-86* to control NSM differentiation. In another neuron class, I3, a homeobox gene *ceh-2* has alone also merely limited effects, but synergized with *ceh-34*. In yet other neuron classes, the collaborating homeobox gene has as strong defects as the *ceh-34* mutation has. The role of these additional homeobox genes as collaborators of *ceh-34* is further corroborated by the enrichment of binding sites for these cofactors (in addition to *ceh-34*) in the gene batteries of the individual neuron classes (as brought up in the Discussion).

Taken together, it is clear that the cofactor analysis is just at its beginning. Our sole purpose here was to show that *ceh-34* does interact with distinct homeobox genes in distinct pharyngeal neuron classes. More analysis clearly is required to fully define the complete set of interacting transcription factors. Given the amount of data already present in the manuscript, we feel that more analysis of these cofactors should be left to future studies.

3) On the same topic, there is no discussion of what complete loss of a marker versus "dimming" means. Specifically, if two TFs are acting in a combinatorial code to dictate a specific neuronal cell-fate, possibly acting cooperatively on the same set of downstream genes (enhancers), but one shows complete loss of marker expression and the other "dimming", what does this mean with regards to TFs-enhancer logic?

This is an excellent question. “Dimming” vs complete expression elimination is something that we have observed – and to a good extent also explained – in the context of transcription factor mutant analysis in other cellular contexts. In some particularly well described case (*ttx-3*/*ceh-10* function in AIY neuron or *unc-86*/*mec-3* function in touch receptor neurons), TFs bind DNA cooperatively and loss of either factor alone essentially eliminates target gene expression. In other cases (e.g. dopamine neuron specification), we have observed an alternative logic, a so-called billboard logic, where TFs bind independently to DNA, and the sum of binding of multiple factors adds up to achieve full blown transcriptional activation; in such cases removal of individual factors results in partially expressive and or partially penetrant effects.

In the revised version, we now explain these issues in the discussion (p.21).

4) They only test the necessity of the genes. While this is certainly the most important issue, many genes may be necessary for a cell's differentiation without really governing its fate. For instance, eye-specific mutant screens in *Drosophila* identified some 9% of all genes as being necessary for eye development (~1,500 genes). However, only a few of these can act to dictate eye development. The study would be greatly strengthened by single and combinatorial misexpression experiments, to probe if these TF codes are also sufficient to dictate cell fate in other cells. Combinatorial misexpression has been extensively done in *Drosophila*, zebrafish, chick, mouse and iPSC, with stellar results, and the worm is lagging behind in this space.

We have now done several ectopic expression experiments: We have co-misexpressed two homeobox genes (*unc-86* and *ttx-3*) that we know to be required for NSM differentiation throughout the pharyngeal nervous system (that already expresses *ceh-34* and its cofactor *eya-1*) and found that each one of three transgenic lines displayed 10-40% penetrant NSM marker ectopic expression but only in two of the 14 pharyngeal neuron classes. Moreover, we have misexpressed the same two NSM regulators (*unc-86*, *ttx-3*) together with *ceh-34* and its cofactor *eya-1* outside the pharyngeal nervous system in a diverse set of ~8 neuron classes (using the *unc-47* driver) and found exceptionally low penetrant effect in two lines (1 animal out of ~50 scored, in two lines).

These very limited defects are entirely expected, based on two confounding factors: (1) In contrast to necessity experiments, where removal of single factors can and does result in differentiation defects, sufficiency require the misexpression of the COMPLETE set of TFs involved in identity specification. We apparently do not know the complete set of regulators for all pharyngeal neurons. (2) As importantly, ectopically expressed TFs need to override the endogenous differentiation program of a neuron. As we have explicitly shown in a paper in *eLife* in 2017 (Patel and Hobert), terminal differentiation does not only involve the activation of a specific gene battery, but also involves the active repression of alternative differentiation programs, likely via chromatin-based mechanisms. Hence, misexpression of a (likely insufficient) combination of TFs is unlikely to result in strong effects. We do mention these negative results and their interpretation now in the Discussion, as requested by the reviewer.

[Editors’ note: further revisions were suggested prior to acceptance, as described below.]

The original reviewers have seen and discussed your responses to the initial reviews. All reviewers agree that you have addressed the majority of their concerns and that the paper is nearly ready for publication. However, there are two remaining issues we would like you to address.1) The reviewers appreciate the detailed rationale you have provided regarding the use of the term "enteric." They largely agree with your point that it is "justified to call the pharyngeal nervous system an enteric nervous system." However, in your paper, and particularly in the title, you refer to this as "the enteric nervous system" of *C. elegans*. This seems to imply that DVB and AVL, which innervate enteric muscles, should not be considered enteric. Is there a way for you to make the point that the pharyngeal nervous system should be considered part of the enteric nervous system without implying that it is its sole component?

We have changed the title to “The enteric nervous system of the *C. elegans* pharynx is specified…” to indicate that we are not dealing with AVL and DVB.

2) Regarding the new misexpression experiments, the reviewers find the results of these studies interesting and feel that they will be useful to others in the field. Some of the reviewers find it surprising that you observed limited effects in these experiments, since in other systems, researchers have obtained robust ectopic generation of various neuronal sub-types by co-misexpression of 2-4 TFs. Further, one reviewer notes that studies in other systems have found that "TF co-misexpression can simply add ectopic neurotransmitter expression on top on the already existing one, hence creating a mixed cell fate," which means that it is not always necessary to "override the endogenous differentiation program," as may be the case in your studies. To address these points, the reviewers would like to you include the new misexpression data in the manuscript, perhaps in an additional supplementary figure, and discuss this issue more thoroughly in the Discussion.

As requested, we have now included the ectopic expression data as a new Supplemental Figure on p.22. As you can see, there are some, albeit limited effects, altogether not too different from is observed in other organisms (where such ectopic expression experiments are usually limited to very few neurons, while we test effects here in *all* pharyngeal neurons). The accompanying text that discusses these results is:

“We found that the ectopic expression of pharyngeal homeobox genes reveal a limited capacity to respecify identity features of pharyngeal neuron (Figure 11 – Supplement 2). This is a likely reflection of our incomplete knowledge of the entire set of collaborating factors and possibly also a reflection of the difficulties associated with overriding endogenous terminal differentiation programs by ectopic expression of drivers of alternative fates (Patel and Hobert, 2017).”

We think there is not much more to discuss here.